# Beyond 2D Representation: Learning 3D Scene Field for Robust Self-supervised Monocular Depth Estimation

## Abstract

Monocular depth estimation has been extensively studied over the past few decades, yet achieving robust depth estimation in real-world scenes remains a challenge, particularly in the presence of reflections, shadow occlusions, and low-texture regions. Existing methods typically rely on extracting front-view 2D features for depth estimation, which often fail to capture those complex physical factors present in real-world scenes, leading to discontinuous, incomplete, or inconsistent depth maps. To address these issues, we turn to learning a more powerful 3D representation for robust monocular depth estimation, and propose a novel self-supervised monocular depth estimation framework based on the **T**hree-dimensional **S**cene **F**ield representation, or TSF-Depth for short. Specifically, we build our TSF-Depth framework upon an encoder-decoder architecture. The encoder extracts scene features from the input 2D image, and subsequently reshapes it as a tri-plane feature field by incorporating scene prior encoding. This tri-plane feature field is designed to implicitly model the structure and appearance of the continuous 3D scene. We then estimate a high-quality depth map from the tri-plane feature field by simulating the camera imaging process. To do this, we construct a 2D feature map with 3D geometry by sampling from the tri-plane feature field using the coordinates of points where the line of sight intersects with the scene. The aggregated multi-view geometric features are subsequently fed into the decoder for depth estimation. Extensive experiments on KITTI and NYUv2 datasets show that TSF-Depth achieves state-of-the-art performance. We also validate the generalization capability of our model on Make3D and ScanNet datasets.

## 1 Introduction

Monocular depth estimation is an essential computer vision task and has wide applications in autonomous driving (Geiger et al., 2013; Menze & Geiger, 2015), robot navigation (Dudek & Jenkin, 2024), and 3D reconstruction (Lyu et al., 2023; Yu et al., 2022), etc. This task aims to infer the depth of each pixel in a single image, thereby recovering the 3D scene structure. Yet, estimating depth from a single image is indeed ill-posed and inherently ambiguous, since a 3D scene can be back-projected from an infinite number of 2D images (Shao et al., 2023). Thus, the lack of sufficient 3D geometric cues in a 2D image poses a substantial challenge for monocular depth estimation.

Early monocular depth estimation (Yuan et al., 2022; Liu et al., 2023; Shao et al., 2024) worked in a supervised manner and yielded relatively accurate depth. However, depth labels are expensive to obtain. Moreover, existing physical devices can only capture sparse scene depth (Moon et al., 2023). Consequently, the sparse supervision and data scale hinder their application in practical scenarios.

Recently, self-supervised monocular depth estimation (Zhang et al., 2023a; Han et al., 2023) has attracted widespread attention. The core of such approaches is to synthesize a 2D image using the estimated depth map and minimize the photometric loss between the synthesized image and the target image (Zhao et al., 2023a). Previous efforts focused on mining effective 2D features for depth estimation by designing advanced network architectures (Lyu et al., 2021; Zhang et al., 2023a), developing more suitable loss functions (Godard et al., 2019; Liu et al., 2024), using semantic information (Casser et al., 2019), or leveraging geometric priors (Zhao et al., 2024; Sun et al., 2024).

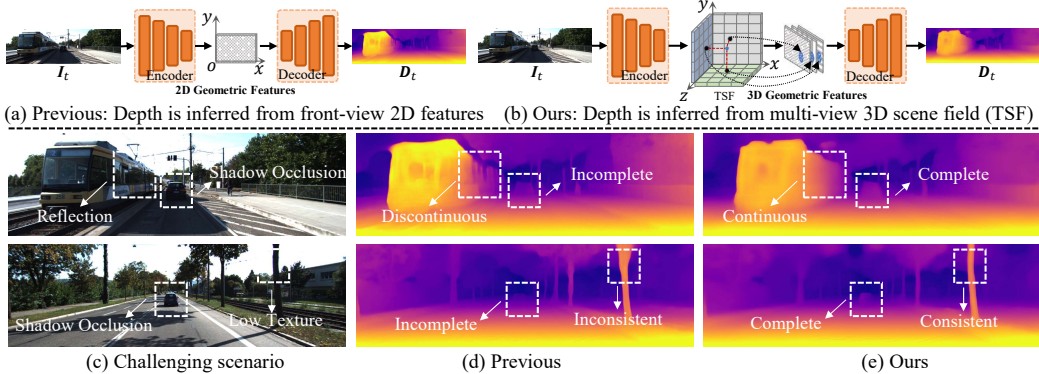

(a) Previous: Depth is inferred from front-view 2D features    (b) Ours: Depth is inferred from multi-view 3D scene field (TSF)

(c) Challenging scenario          (d) Previous          (e) Ours

Figure 1: **An illustration of our motivation**. Real-world 3D scenes usually contain numerous challenges, including reflections, shadow occlusions, low textures and so on, which causes existing methods (*e.g.*, Lite-Mono-8M (Zhang et al., 2023a)) that typically rely on front-view 2D features to obtain discontinuous, incomplete, or inconsistent depth maps. To this end, we propose to model a multi-view 3D scene filed, thereby capturing 3D geometric features for robust depth estimation.

Although these methods have shown satisfactory performance in conventional scenes, yet achieving robust depth estimation in real-world scenes remains a challenge, particularly in the presence of reflections, shadow occlusions, and low-texture regions (see Fig. 1 (c)). The main challenge posed by such scenarios is that these local regions often lack sufficient discriminative depth cues. In response, humans simulate a roughly 3D scene corresponding to the 2D image and then combine geometry cues from horizontal, vertical, and depth direction to infer the depth of a particular pixel. Even if information is lacking in one direction, geometric clues from other directions can supplement it. However, almost all existing methods typically rely on extracting front-view 2D features for depth estimation, which often fail to capture those complex physical factors present in real-world scenes. Thus, as shown in Fig. 1 (a) and (d), existing methods are limited by the paradigm of the front-view 2D representation, which often do not contain sufficient 3D geometric cues (Han et al., 2023), resulting in discontinuous, incomplete, or inconsistent depth maps.

In this paper, we propose a novel self-supervised monocular depth estimation framework based on the **T**hree-dimensional **S**cene **F**ield representation, or TSF-Depth for short. Unlike previous methods that employ only the front-view 2D features, we design a 3D scene field to recover the multi-view representation and then capture sufficient structure- and orientation-aware 3D geometric features from it for robust depth estimation (see Fig. 1 (b) and (e)). Specifically, we build our TSF-Depth upon an encoder-decoder architecture. The encoder extracts scene features from the input 2D image, and then are reshaped as a tri-plane feature field with three axis-aligned orthogonal feature planes by incorporating scene prior encoding. This tri-plane feature field is designed to implicitly model the structure and appearance of the continuous 3D scene. We then estimate a high-quality depth map from the tri-plane feature field by simulating the camera imaging process. To achieve this, we construct a 2D feature map with 3D geometry by sampling from the tri-plane feature field using the coordinates of the points where the line of sight intersects with the scene. The aggregated multi-view geometric feature map is fed into the decoder for depth estimation. Extensive experiments on four datasets validate the state-of-the-art and generalization capabilities of TSF-Depth.

To summarize, the main contributions of our work are as follows:

- We propose a novel self-supervised monocular depth estimation framework based on the **T**hree-dimensional **S**cene **F**ield representation (TSF-Depth). To the best of our knowledge, our TSF-Depth is the first work to model 3D scene filed for monocular depth estimation.

- We design a tri-plane feature field that is reshaped from hybrid features of scene content and scene prior encoding to model the multi-view representation of the continuous 3D scene.

- We attentively design a 3D-to-2D mapping strategy to sample 2D features with 3D geometry from tri-plane feature field by simulating camera image, *i.e.*, projecting the coordinates of the points where the line of sight intersects with the scene onto three orthogonal planes.

- Extensive experiments on widely used outdoor datasets (KITTI and Make3D) and indoor datasets (NYUv2 and ScanNet) show the robustness and generalization capabilities.

## 2 RELATED WORK

**Supervised Monocular Depth Estimation.** Eigen et al. (2014) first used a coarse-to-fine network for monocular depth estimation. Subsequently, numerous supervised works have been proposed. These works can be functionally classified into regression-based methods (Ranftl et al., 2021; Zhao et al., 2021; Shao et al., 2023) and classification-based methods (Bhat et al., 2021; Hu et al., 2022; Shao et al., 2024). Regression-based works use convolutional neural networks to directly learn the depth value of each pixel by minimizing the error between the prediction and ground-truth depths. However, these methods usually suffer from slow convergence and local solutions (He et al., 2022). Classification-based methods divide the depth range into different bins, and predict the probability of falling in each bin to obtain the final depth by weighted summation, which is easier to optimize. However, the high cost of data collection for training limits the wide application of these methods.

**Self-Supervised Monocular Depth Estimation.** Self-supervised depth estimation approaches that avoid the need for ground-truth depth during training phase have gained attention. Zhou et al. (2017) proposed a pioneering work that utilized depth network and pose network to jointly estimate depth map and camera pose, and only adopted monocular video as training data. Following this classical joint training pipeline, subsequent works improve the performance by designing robust losses (Gordon et al., 2019; Shu et al., 2020; Zhan et al., 2018), using auxiliary information during training (Watson et al., 2019; Klodt & Vedaldi, 2018), dealing with moving objects (Godard et al., 2019; Klingner et al., 2020), and adding extra geometric constraints (Yang et al., 2018; Li et al., 2021). Yet, these methods also have limitations as they infer depth from the fron-view 2D feature space, which often do not contain sufficient 3D geometric cues (Han et al., 2023), and ignore the value of additional scene geometry priors in depth estimation. Instead, our TSF-Depth models 3D scene field using scene feature and scene priors to recover the multi-view representation and then capture sufficient 3D geometric features from it for robust depth estimation.

**3D Scene Representation.** Depth estimation using only 2D representation is a well-known ill-posed problem. To this end, learning-based multi-view stereo (MVS) methods (Yao et al., 2018; 2019; Yang et al., 2022) use the cost volume as spatial representation of the scene and then utilize 3D CNNs to continuously learn full-space 3D features. Considering the point cloud structure makes 3D feature learning more flexible compared to the cost volume, some point cloud-based MVS (Chen et al., 2019; 2020; Zhao et al., 2023b) propose to replace the cost volume with the point cloud as the spatial representation of the scene. Although these methods establish the spatial structure feature of the scene or learn the global feature of the scene, the structural attributes are not further perceived and learned. Additionally, 3D CNNs require memory cubic to the model resolution, which can be a hindrance to achieving optimal performance. Unlike existing MVS methods require complex cost volume, multi-view images and 3D CNNs, TSF-Depth models a single-view image into a tri-plane feature field as a 3D scene representation using only a 2D encoder-decoder architecture.

## 3 METHOD

### 3.1 OVERVIEW

Previous studies relied on extracting front-view 2D features for depth estimation. Although these methods are effective in conventional scenes, as discussed in Section 1, they often fail to represent those complex physical factors present in real-world scenes due to the limitations of front-view 2D representations. To this end, we propose TSF-Depth, a novel self-supervised depth estimation framework based on the 3D scene representation. The TSF-Depth is designed to model a multi-view representation of the continuous 3D scene with a tri-plane feature field. Then, we can achieve robust depth estimation using 2D features with 3D geometry sampled from the 3D scene field.

The proposed pipeline, illustrated in Fig. 2, consists of two essential steps: modeling 3D scene with tri-plane feature field and depth estimation with 3D scene field. Give a target image $\boldsymbol{I}_t \in \mathbb{R}^{H \times W \times 3}$, and image coordinates $\{\boldsymbol{C}^s \in \mathbb{R}^{H/2^s \times W/2^s \times 3}\}_{s=1}^{S}$ at multiple scales, an encoder is used to extract multi-scale scene features $\{\mathbf{F}^s \in \mathbb{R}^{H/2^s \times W/2^s \times C}\}_{s=1}^{S}$ from the former, and positional encoding is performed to obtain multi-scale scene priors $\{\boldsymbol{E}^s \in \mathbb{R}^{H/2^s \times W/2^s \times C}\}_{s=1}^{S}$ from the latter. The $H$ and $W$ denote the height and width of image, while $S$ and $C$ denotes the feature scale and channel. The scene priors not only initialize 3D scene structure, but also provide spatial cues. Then, the scene

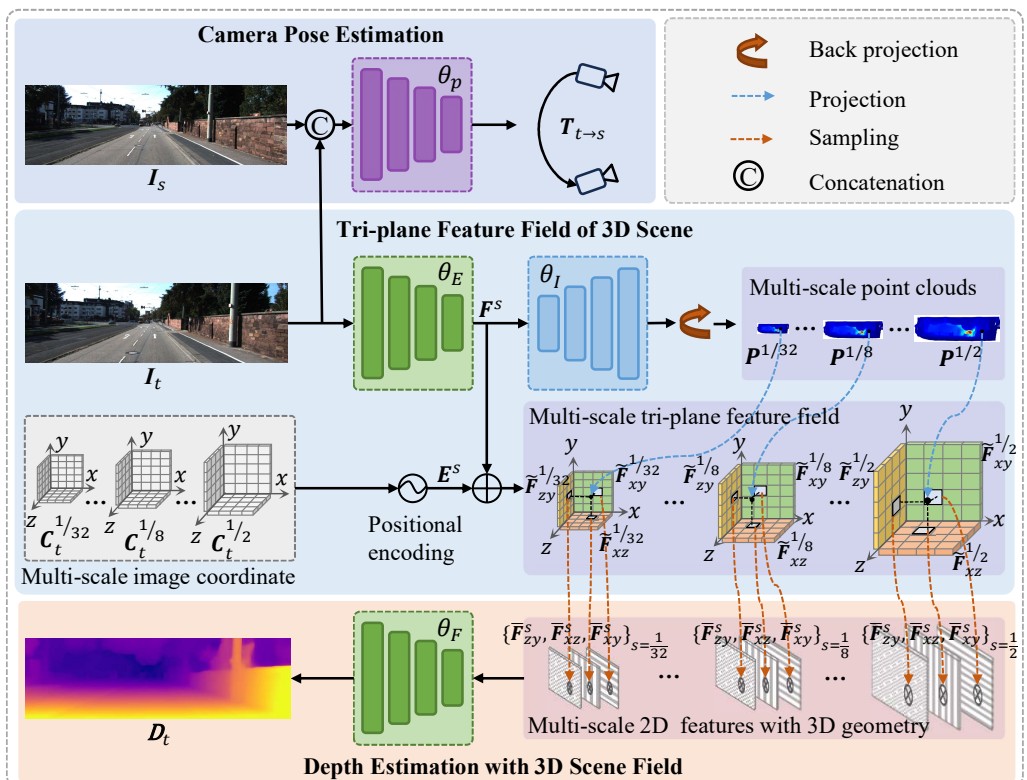

Figure 2: **Overview of the proposed TSF-Depth.** Given a target image and image coordinates at multiple scales, the scene features and scene prior features are first extracted and summed to generate multi-scale hybrid features. The hybrid features are reshaped into multi-scale tri-plane feature fields to implicitly model the multi-view representation of the 3D scene. We then recover the coordinates of the points where the line of sight intersects with the scene, and project all points onto three orthogonal planes to retrieve the 2D feature with 3D geometric for robust depth estimation.

features are split into three $\frac{C}{3}$-channel feature maps in the channel dimension, and incorporated into scene priors to generate multi-scale hybrid features $\{\tilde{\boldsymbol{F}}^s \in \mathbb{R}^{H/2^s \times W/2^s \times C}\}_{s=1}^{S}$ with scene structure awareness. Subsequently, the hybrid features are reshaped as axis-aligned orthogonal tri-plane feature field $\{\tilde{\boldsymbol{F}}^s_{xy} \in \mathbb{R}^{H/2^s \times W/2^s \times C/3}, \tilde{\boldsymbol{F}}^s_{xz} \in \mathbb{R}^{H/2^s \times W/2^s \times C/3}, \tilde{\boldsymbol{F}}^s_{zy} \in \mathbb{R}^{H/2^s \times W/2^s \times C/3}\}_{s=1}^{S}$, thereby implicitly modeling the multi-view representation of the continuous 3D scene. Meanwhile, we again use intermediate semantic features to recover the coordinates of the points where the line of sight from each pixel intersects with the scene. Finally, we project all 3D points $\boldsymbol{P}$ onto three orthogonal feature planes, retrieving the corresponding feature $\{\bar{\boldsymbol{F}}^s_{xy} \in \mathbb{R}^{H/2^s \times W/2^s \times C/3}, \bar{\boldsymbol{F}}^s_{xz} \in \mathbb{R}^{H/2^s \times W/2^s \times C/3}, \bar{\boldsymbol{F}}^s_{zy} \in \mathbb{R}^{H/2^s \times W/2^s \times C/3}\}_{s=1}^{S}$ via bilinear interpolation, and aggregating these multi-view geometric feature map via concatenation to predict the depth map. During training phase, we follow classic self-supervised depth estimation approaches to simultaneously learning a PoseNet to predict relative pose, which will be combined with depth to construct the optimized object.

## 3.2 TRI-PLANE FEATURE FIELD OF 3D SCENE

Inspired by humans combine depth cues from horizontal, vertical, and depth directions to infer the depth of a particular pixel, we thus implicitly model a three-plane feature field to recover the multi-view representation. In addition, due to the ill-posed depth estimation task, relying solely on image content to infer depth is limited. Observing that the pixel depth in an image is closely related to its relative spatial position, we thus explore this regularity and incorporate it into our 3D scene Field.

**Scene Feature Extracting.** Given target image $\boldsymbol{I}_t \in \mathbb{R}^{H \times W \times 3}$, we employ a 2D encoder to capture multi-scale secene features:

$$\boldsymbol{F}^s = \theta_E(\boldsymbol{I}_t), \tag{1}$$

where $\boldsymbol{F}^s \in \mathbb{R}^{H/2^s \times W/2^s \times C}$. Benefiting from the powerful learning and representation capabilities of neural networks, it is feasible to learn features of 3D scenes in different directions from the input image. To this end, we split the semantic feature maps in the channel dimension and form three $\frac{C}{3}$-channel feature maps $\{\boldsymbol{F}_{xy}^s \in \mathbb{R}^{H/2^s \times W/2^s \times \frac{C}{3}}, \boldsymbol{F}_{xz}^s \in \mathbb{R}^{H/2^s \times W/2^s \times \frac{C}{3}}, \boldsymbol{F}_{yz}^s \in \mathbb{R}^{H/2^s \times W/2^s \times \frac{C}{3}}\}_s^S$ as the preliminary representation of the three orthogonal views of the 3D scene.

**Scene Prior Encoding.** In order to reasonably incorporate the scene prior, *i.e.*, relative spatial position, instead of directly passing the image coordinate into network, we introduce a positional encoding to map the image coordinate to a high-dimensional feature vector. More details about scene prior are discussed in Appendix B. Formally, the position encoding function is defined as:

$$\gamma(c) = (\sin(2^0 \pi c), \cos(2^0 \pi c), \cdots, \sin(2^{L-1} \pi c), \cos(2^{L-1} \pi c)), \tag{2}$$

where $c$ is the stored value of coordinate, $L$ is the number of encoding frequencies, and $\gamma(c)$ denotes the mapping of $c$ from $\mathbb{R}$ into a higher dimensional space $\mathbb{R}^{2L}$. Thus, given the homogeneous coordinates $\{\boldsymbol{C}_{xy}^s \in \mathbb{R}^{H/2^s \times W/2^s \times 3}, \boldsymbol{C}_{xz}^s \in \mathbb{R}^{H/2^s \times W/2^s \times 3}, \boldsymbol{C}_{zy}^s \in \mathbb{R}^{H/2^s \times W/2^s \times 3}\}$ of three orthogonal views at multiple scales, the multi-scale and multi-view scene prior can be obtained by:

$$\begin{cases} \boldsymbol{E}_{xy}^s(u, v) = \Xi(\gamma(u'), \gamma(v'), \gamma(1)), \\ \boldsymbol{E}_{xz}^s(u, v) = \Xi(\gamma(u'), \gamma(1), \gamma(v')), \\ \boldsymbol{E}_{zy}^s(u, v) = \Xi(\gamma(1), \gamma(v'), \gamma(u')). \end{cases} \tag{3}$$

where $u' = \frac{2u}{W/2^s - 1} - 1$, $v' = \frac{2v}{W/2^s - 1} - 1$, and $\Xi[\cdot]$ is the concatenation operator. Here, $u'$ and $v'$ are normalized to $[-1, 1]$, which ensure scale consistency of scene prior and numerical stability.

Since the channel dimension of the generated scene prior features is controlled by $L$, it may not match the scene feature. We further introduce two $1 \times 1$ convolutions to adjust the channel dimension of as $\{\boldsymbol{E}_{xy}^s \in \mathbb{R}^{H/2^s \times W/2^s \times C/3}, \boldsymbol{E}_{xy}^s \in \mathbb{R}^{H/2^s \times W/2^s \times C/3}, \boldsymbol{E}_{xy}^s \in \mathbb{R}^{H/2^s \times W/2^s \times C/3}\}_{s=1}^S$.

**Multi-scale Tri-plane Feature Field.** After the above two steps, we obtain the scene features that represent the semantic details of 3D scene, and the scene prior features that initialize 3D scene structure, but also provide spatial cues. Subsequently, we incorporate the scene semantic features into scene prior features to obtain multi-scale hybrid features with scene structure awareness:

$$\begin{cases} \tilde{\boldsymbol{F}}_{xy}^s = \boldsymbol{F}_{xy}^s + \boldsymbol{E}_{xy}^s, \\ \tilde{\boldsymbol{F}}_{xz}^s = \boldsymbol{F}_{xz}^s + \boldsymbol{E}_{xz}^s, \\ \tilde{\boldsymbol{F}}_{zy}^s = \boldsymbol{F}_{zy}^s + \boldsymbol{E}_{zy}^s. \end{cases} \tag{4}$$

The three hybrid feature planes are axis-aligned orthogonal planes, which are defined as our tri-plane feature fields. The $\tilde{\boldsymbol{F}}_{xy}^s$ perceives the continuous change of depth in z-direction, the $\tilde{\boldsymbol{F}}_{xz}^s$ perceives the consistency of depth in the vertical direction and the $\tilde{\boldsymbol{F}}_{zy}^s$ perceives the similarity of depth in the horizontal direction. Therefore, this tri-plane feature field is designed to implicitly model the structure, orientation, appearance of the continuous 3D scene. Moreover, benefiting from the design of our multi-scale and multi-view tri-plane feature fields, our method can effectively perceive the 3D scene scale, thereby alleviating the ambiguity of monocular depth estimation.

## 3.3 Depth Estimation with 3D Scene Field

After model the 3D scene filed with tri-plane feature field, we aim to capture the 2D feature with 3D geometric from it for depth estimation. However, for each pixel in the image space, we do not know its corresponding position in the 3D scene field. To address this issue, we simulating the inverse process of camera imaging, *i.e.*, we need to recover the coordinates of the points where the line of sight intersects the scene. In other words, we need to estimate the point clouds of the input image.

Point clouds are often used as structural representations of 3D scenes. Many works use 3D CNNs to directly estimate 3D point clouds, but they are costly in terms of efficiency and storage. In this work, we employ the depth map as intermediate representation to efficiently reconstruct the point clouds. To this end, we use a decoder $\theta_I$ to predict the initial depth map $\boldsymbol{D}_{I,t}^s = \theta_I(\Xi[\boldsymbol{F}_{xy}^s, \boldsymbol{F}_{xz}^s, \boldsymbol{F}_{zy}^s])$.

After the above step, we can obtain multi-scales depth maps $\boldsymbol{D}_{I,t}^s \in \mathbb{R}^{\frac{H}{2^s} \times \frac{W}{2^s}}$. Since the scale and shift coefficients of the predicted depth maps are unknown, the reconstructed 3D structure from them is likely to be distorted from inappropriate affine changes. We thus use all scales of depth map to reconstruct multi-scale point clouds from coarse to fine. Given the estimated initial depth map $\boldsymbol{D}_{I,t}^s$ with scale $s$ and camera intrinsics, we can reconstruct the point cloud from the pixel coordinate

based on the pinhole camera model. Specifically, for a 2D point $\boldsymbol{p}_i = [u, v]^T$ in the pixel coordinate system, it can be reprojected back to a 3D point $\boldsymbol{P}_i = [X, Y, Z]^T$ in the camera system by:

$$[X, Y, Z]^T = \boldsymbol{D}_{I,t}^s(u, v)\boldsymbol{K}^{-1}[u, v, 1]^T, \tag{5}$$

where $\boldsymbol{K} \in \mathbb{R}^{3 \times 3}$ denotes the camera intrinsic. Therefore, using Eq. 5, we can explicitly reconstruct multi-scale point clouds $\{\boldsymbol{P}^s \in \mathbb{R}^{\frac{H}{2^s} \times \frac{W}{2^s} \times 3}\}_s^S$. It should be noted that for different scale depths, the coordinate range will also change accordingly, *i.e.*, $u \in \left[0, \frac{W}{2^s} - 1\right]$, $v \in \left[0, \frac{H}{2^s} - 1\right]$.

After the implicitly and explicitly modeling 3D scene phase, we obtain the dense tri-plane feature field and sparse point clouds, respectively. Although the former learns from the 2D image, they have the ability to perceive the 3D scene structure and provide direction-aware multi-view features. The later learn from 2D feature space and lacks of 3D scene awareness, but they can provide approximate spatial structure. To obtain the final precise and robust depth prediction, we design a 3D-to-2D mapping strategy to sample 2D features with 3D geometry by combining the advantage of both.

Since there are different spatial scales between each 3D point in the point cloud and each orthogonal planes of tri-plane feature fields, they need to be aligned in the same space. Compare to back-projecting the each orthogonal planes of tri-plane feature field from the 2D space to 3D space, projecting point cloud from 3D space to 2D space is more efficient. With the multi-scale point cloud and tri-plane feature field, we perform an orthographic projection onto the three axis-aligned planes:

$$[x, y, z]^T = \boldsymbol{K}[X, Y, Z]^T. \tag{6}$$

Generally, the value ranges of point $[x, y, z]$ in three axis are different, *i.e.*, $x \in [0, W - 1]$, $y \in [0, H - 1]$ and $z \in [0, M]$, where $M$ is the maximum depth. To ensure that the projected 2D points are aligned with each planes without going out of bounds, they need to be normalized to $[-1, 1]$:

$$\begin{cases} \bar{x} = (x/(W-1) - 0.5) \times 2, \\ \bar{y} = (x/(H-1) - 0.5) \times 2, \\ \bar{z} = (z/M - 0.5) \times 2. \end{cases} \tag{7}$$

Ultimately, we obtain the 2D points of the point cloud projected onto each orthogonal plane, *i.e.*, the $p_{xy}^s = [\bar{x}, \bar{y}]$ is the projected 2D point located at $\tilde{\boldsymbol{F}}_{xy}^s$, the $p_{xz}^s = [\bar{x}, \bar{z}]$ is the projected 2D point located at $\tilde{\boldsymbol{F}}_{xz}^s$ and the $p_{zy}^s = [\bar{z}, \bar{y}]$ is the projected 2D point located at $\tilde{\boldsymbol{F}}_{zy}^s$. Then, we sample 2D feature map with 3D geometric from each orthogonal plane of tri-planes feature fields using differentiable bilinear sampling operator:

$$\begin{cases} \bar{\boldsymbol{F}}_{xy}^s = \tilde{\boldsymbol{F}}_{xy}^s \langle p_{xy}^s \rangle, \\ \bar{\boldsymbol{F}}_{xz}^s = \tilde{\boldsymbol{F}}_{xz}^s \langle p_{xz}^s \rangle, \\ \bar{\boldsymbol{F}}_{zy}^s = \tilde{\boldsymbol{F}}_{zy}^s \langle p_{zy}^s \rangle. \end{cases} \tag{8}$$

where $\langle \cdot \rangle$ is the sampling operator (Jaderberg et al., 2015). Finally, the multi-view geometric features aggregated through concatenation, and a depth decoder is used to predict the high-quality depth map:

$$\boldsymbol{D}_{F,t} = \theta_F(\Xi[\bar{\boldsymbol{F}}_{xy}, \bar{\boldsymbol{F}}_{xz}, \bar{\boldsymbol{F}}_{zy}]). \tag{9}$$

### 3.4 SELF-SUPERVISED LEARNING

Following monodepth2 (Godard et al., 2019), we use a target frame $\boldsymbol{I}_t$ and two adjacent frames $\boldsymbol{I}_a$ ($a \in \{t-1, t+1\}$) to jointly train a DepthNet ($\theta_E$, $\theta_I$, $\theta_F$) and a PoseNet $\theta_p$. During training, $\boldsymbol{I}_t$ is fed into the DepthNet to get the depth $\boldsymbol{D}_{F,t}$, and $(\boldsymbol{I}_t, \boldsymbol{I}_a)$ are put into PoseNet to get the relative camera pose $\boldsymbol{T}_{t \to a}$. Then, we can synthesize a target frame $\boldsymbol{I}_{F,a \to t}$ by warping the source frame $\boldsymbol{I}_t$: $\boldsymbol{I}_{F,a \to t} = \boldsymbol{I}_a \langle proj(\boldsymbol{D}_{F,t}, \boldsymbol{T}_{t \to a}, \boldsymbol{K}) \rangle$, where $proj(\cdot)$ is the coordinate projection operation (Zhou et al., 2017). The photometric error between $\boldsymbol{I}_{F,t}$ and $\boldsymbol{I}_t$, consisting of $L_1$ and $SSIM$ weighted by $\alpha$, is calculated as: $pe(\boldsymbol{I}_t, \boldsymbol{I}_{F,a \to t}) = \frac{\alpha}{2}(1 - SSIM(\boldsymbol{I}_t, \boldsymbol{I}_{F,a \to t})) + (1 - \alpha)\|\boldsymbol{I}_t - \boldsymbol{I}_{F,a \to t}\|_1$. Following Monodepth2, we adopt the per-pixel minimum photometric loss as our reprojection loss:

$$\mathcal{L}_{ph}(\boldsymbol{I}_t, \boldsymbol{I}_{F,a \to t}) = \min_a pe(\boldsymbol{I}_t, \boldsymbol{I}_{F,a \to t}). \tag{10}$$

We also use the edge-aware smoothness loss to encourage locally smooth depth maps:

$$\mathcal{L}_{sm}(\boldsymbol{D}_{F,t}, \boldsymbol{I}_t) = \left|\partial_x \boldsymbol{D}_{F,t}^*\right| e^{-|\partial_x \boldsymbol{I}_t|} + \left|\partial_y \boldsymbol{D}_{F,t}^*\right| e^{-|\partial_y \boldsymbol{I}_t|}. \tag{11}$$

where $\partial_x$ and $\partial_y$ are the gradients in the horizontal and vertical direction respectively. Besides, $\boldsymbol{D}_{F,t}^* = \boldsymbol{D}_{F,t}/\bar{\boldsymbol{D}}_{F,t}$ is the mean-normalized inverse depth from Monodepth2 to discourage shrinking of the estimated depth. Therefore, the total loss for final depth map is defined as:

$$\mathcal{L}_F = \beta \mathcal{L}_{ph}(\boldsymbol{I}_t, \boldsymbol{I}_{F,a \to t}) + \gamma \mathcal{L}_{sm}(\boldsymbol{D}_{F,t}, \boldsymbol{I}_t). \tag{12}$$

Table 1: **Depth estimation results on KITTI** (Geiger et al., 2013). We divide compared methods into three categories. S: stereo training. M: monocular training. MS: stereo and monocular training.

| Method | Train | Error Metric (↓) | | | | Accuracy Metric (↑) | | |
|---|---|---|---|---|---|---|---|---|
| | | Sq Rel | Abs Rel | RMSE | RMSE log | $\delta<1.25$ | $\delta<1.25^2$ | $\delta<1.25^3$ |
| Monodepth (Godard et al., 2017) | S | 1.344 | 0.148 | 5.927 | 0.247 | 0.803 | 0.922 | 0.964 |
| 3Net (Poggi et al., 2018) | S | 1.201 | 0.119 | 5.888 | 0.208 | 0.844 | 0.941 | 0.978 |
| Monodepth2 (Godard et al., 2019) | S | 0.873 | 0.109 | 4.960 | 0.208 | 0.864 | 0.948 | 0.975 |
| DepthHints (Watson et al., 2019) | S | 0.780 | 0.106 | 4.695 | 0.193 | 0.875 | 0.958 | 0.980 |
| BRNet (Han et al., 2022) | S | 0.792 | 0.103 | 4.716 | 0.197 | 0.876 | 0.954 | 0.978 |
| Monodepth2 (Godard et al., 2019) | MS | 0.818 | 0.106 | 4.750 | 0.196 | 0.874 | 0.957 | 0.979 |
| DepthHints (Watson et al., 2019) | MS | 0.769 | 0.105 | 4.627 | 0.189 | 0.875 | 0.959 | 0.982 |
| HR-Depth (Lyu et al., 2021) | MS | 0.785 | 0.107 | 4.612 | 0.185 | 0.887 | 0.962 | 0.982 |
| R-MSFM6 (Zhou et al., 2021b) | MS | 0.787 | 0.111 | 4.625 | 0.189 | 0.882 | 0.961 | 0.981 |
| GeoNet (Yin & Shi, 2018) | M | 1.060 | 0.149 | 5.567 | 0.226 | 0.796 | 0.935 | 0.975 |
| Monodepth2 (Godard et al., 2019) | M | 0.903 | 0.115 | 4.863 | 0.193 | 0.877 | 0.959 | 0.971 |
| DepthHints (Watson et al., 2019) | M | 0.845 | 0.109 | 4.800 | 0.196 | 0.870 | 0.956 | 0.980 |
| S³Net (Cheng et al., 2020) | M | 0.826 | 0.124 | 4.981 | 0.200 | 0.846 | 0.955 | 0.982 |
| HR-Depth (Lyu et al., 2021) | M | 0.792 | 0.109 | 4.632 | 0.185 | 0.884 | 0.962 | 0.983 |
| CADepth-Net (Yan et al., 2021) | M | 0.769 | 0.105 | 4.535 | 0.181 | 0.892 | 0.964 | 0.983 |
| DIFFNet (Zhou et al., 2021a) | M | 0.764 | 0.102 | 4.483 | 0.180 | 0.896 | 0.965 | 0.983 |
| DynaDepth (Zhang et al., 2022a) | M | 0.761 | 0.108 | 4.608 | 0.187 | 0.883 | 0.962 | 0.982 |
| MonoFormer (Bae et al., 2023) | M | 0.846 | 0.104 | 4.580 | 0.183 | 0.891 | 0.962 | 0.982 |
| SC-DepthV3 (Sun et al., 2023) | M | 0.756 | 0.118 | 4.709 | 0.188 | 0.864 | 0.960 | 0.984 |
| Zhang *et al.* (Zhang et al., 2023b) | M | 0.786 | 0.105 | 4.572 | 0.182 | 0.890 | 0.964 | 0.983 |
| Lite-Mono (Zhang et al., 2023a) | M | 0.765 | 0.107 | 4.561 | 0.183 | 0.886 | 0.963 | 0.983 |
| Lite-Mono-8M (Zhang et al., 2023a) | M | 0.729 | 0.101 | 4.454 | 0.178 | 0.897 | 0.965 | 0.983 |
| Zhao *et al.* Zhao et al. (2024) | M | 0.809 | 0.110 | 4.616 | 0.185 | - | - | - |
| Xiong *et al.* (Xiong et al., 2024) | M | 0.868 | 0.122 | 4.986 | 0.200 | 0.857 | 0.953 | 0.980 |
| ShuffleMono (Feng et al., 2024) | M | 0.850 | 0.114 | 4.821 | 0.193 | 0.872 | 0.957 | 0.980 |
| Liu *et al.* (Liu et al., 2024) | M | 0.747 | 0.114 | 4.724 | 0.187 | 0.863 | 0.960 | 0.984 |
| Dynamo-Depth (Sun et al., 2024) | M | 0.758 | 0.112 | 4.505 | 0.183 | 0.873 | 0.959 | 0.984 |
| **TSF-Depth** | **M** | **0.692** | **0.096** | **4.335** | **0.173** | **0.903** | **0.967** | **0.984** |

To reconstruct accuracy point cloud, we apply a photometric loss to the initial depth:

$$\mathcal{L}_I = \frac{1}{S}\sum_{s=1}^{S}\varphi\mathcal{L}_{ph}(\boldsymbol{D}_{F,t}, \mathbb{U}(\boldsymbol{D}_{I,t}^s)). \tag{13}$$

where $\mathbb{U}$ is the upsampling operation. Our overall loss function can be formulated as:

$$\mathcal{L} = \mathcal{L}_I + \mathcal{L}_F. \tag{14}$$

# 4 EXPERIMENTS

## 4.1 DATASETS AND EVALUATION METRICS

**Outdoor Datasets. KITTI** (Geiger et al., 2013) is an outdoor benchmark with a resolution of $1242 \times 375$. Following Zhou et al. (2017), we use 39810, 4424, and 697 images for training, validation, and testing, respectively. **Make3D** (Saxena et al., 2008) contains 134 outdoor image-depth pairs with a resolution of $1704 \times 2272$, which is used in this work as a generalization test.

**Indoor Datasets. NYUv2** (Silberman et al., 2012) is an indoor benchmark with a resolution of $640 \times 480$. Following StructDepth (Li et al., 2021), we use the official training and validation splits which include 302 and 33 sequences, and use officially provided 654 images for testing. **ScanNet** (Dai et al., 2017) contains 1513 indoor scenes with a resolution of $1296 \times 968$. We use official data split and use this dataset as a generalization test for indoor scenes.

**Evaluation Metrics.** We follow Monodepth2 (Godard et al., 2019) using relative squared error (**Sq Rel**), absolute relative error (**Abs Rel**), root mean squared error (**RMSE**), root mean squared logarithmic error (**RMSE log**) and threshold accuracy ($\sigma<1.25$, $\sigma<1.25^2$ and $\sigma<1.25^3$).

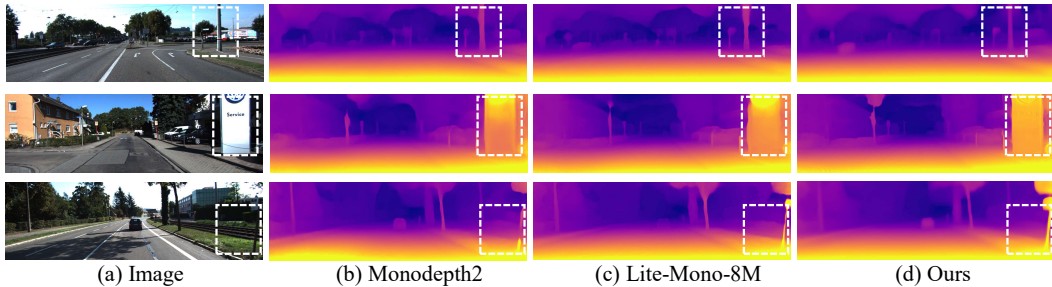

|     (a) Image     |     (b) Monodepth2     |     (c) Lite-Mono-8M     |     (d) Ours     |

Figure 3: **Qualitative comparison on KITTI** (Geiger et al., 2013). We highlight challenging areas.

Table 2: **Generalization on Make3D** (Saxena et al., 2008). All methods are trained on KITTI.

| Method | Error Metric ($\downarrow$) | | | |
|---|---|---|---|---|
| | Sq Rel | Abs Rel | RMSE | RMSE log |
| Monodepth2 (Godard et al., 2019) | 3.589 | 0.322 | 7.418 | 0.163 |
| HR-Depth (Lyu et al., 2021) | 3.208 | 0.315 | 7.024 | 0.159 |
| DIFFNet (Zhou et al., 2021a) | 3.313 | 0.309 | 7.008 | 0.155 |
| DynaDepth (Zhang et al., 2022a) | 3.311 | 0.334 | 7.463 | 0.169 |
| Chen *et al.* (Chen et al., 2023) | 3.610 | 0.370 | 7.133 | - |
| Zhang *et al.* (Zhang et al., 2023b) | 3.485 | 0.314 | 7.188 | - |
| Lite-Mono (Zhang et al., 2023a) | 3.060 | 0.305 | 6.981 | 0.158 |
| Zhao *et al.* (Zhao et al., 2024) | 3.200 | 0.316 | 7.095 | 0.158 |
| Xiong *et al.* (Xiong et al., 2024) | 3.102 | 0.319 | 7.005 | 0.161 |
| **TSF-Depth** | **2.925** | **0.292** | **6.744** | **0.150** |

## 4.2 IMPLEMENTATION DETAILS

We implement TSF-Depth in PyTorch (Paszke et al., 2017), traning it for 20 epochs on the outdoor dataset and 40 epochs on the indoor dataset by using AdamW (Loshchilov & Hutter, 2017) optimizer on a single RTX 3090 GPU. The batch size is set to 12 for the outdoor dataset and 16 for the indoor dataset. The initial learning rate for PoseNet and depth decoder is $1 \times 10^{-4}$, while the depth encoder is trained with an initial learning rate of $5 \times 10^{-5}$. For the PoseNet, we use the same architecture as Monodepth2 (Godard et al., 2019). For the encoder $\theta_E$, initial depth decoder $\theta_I$ and final depth decoder $\theta_F$ of DepthNet, we choose mpvit, the decoder of Monodepth2 (Godard et al., 2019) and HRDecoder (He et al., 2022), respectively. The hyper-parameters $S$, $L$, $\alpha$, $\beta$, $\gamma$, and $\varphi$ are set to 5, 10, 0.85, 1.0, 0.001, and 0.5 respectively. More implementation details are reported in Appendix C.

## 4.3 COMPARISON ON OUTDOOR SCENE

**KITTI.** Table 1 presents the quantitative comparison at resolution of $640 \times 192$ on the outdoor benchmark, *i.e.*, KITTI dataset (Geiger et al., 2013) . Compared to existing methods trained on monocular videos, our method outperforms all these works by significant margins, and also outperforms counterparts trained with additional stereo pairs. In particular, our method relatively outperforms the SOTA method Lite-Mono-8M by $5.1\%$ and by $5.0\%$ in terms of Sq Rel and Abs Rel, respectively. We also compare the qualitative performance with the classic work Monodepth2 (Godard et al., 2019) and the SOTA work Lite-Mono-8M (Zhang et al., 2023a). Fig. 3 presents three visual samples and highlights challenging areas. We observed that for the first two examples, the traditional CNN-based MonoDepth2 and the attention-based Lite-Mono-8M, which extract only front-view 2D feature for depth estimation, both obtain inconsistent depth map. For the third challenging example with low texture, the compared methods obtained discontinuous results. In contrast, our method generates superior visual results due to our multi-view representation to capture the 3D geometric cues for robust depth estimation. More visualization results are shown in Appendix D. In addition, the quantitative results at $1280 \times 384$ and $1024 \times 320$ resolutions are reported in Appendix E.

**Make3D.** To show the generalization capability, we further test our proposed method on the Make3D dataset (Saxena et al., 2008). Following the evaluation strategy in (Zhou et al., 2017), our model is training on the KITTI dataset (Geiger et al., 2013) without any fine-tuning on the Make3D dataset. As shown in Table 2, our method outperforms all other self-supervised monocular depth estimation approaches, which demonstrates our models can be well generalized to unseen outdoor scenes.

Table 3: **Depth estimation results on NYUv2** (Silberman et al., 2012).

| Method | Error Metric (↓) | | Accuracy Metric (↑) | | |
|---|---|---|---|---|---|
| | Abs Rel | RMSE | $\delta < 1.25$ | $\delta < 1.25^2$ | $\delta < 1.25^3$ |
| MovingIndoor (Zhou et al., 2019) | 0.208 | 0.712 | 0.674 | 0.900 | 0.968 |
| Monodepth2 (Godard et al., 2019) | 0.160 | 0.601 | 0.767 | 0.949 | 0.988 |
| TrainFlow (Zhao et al., 2020) | 0.189 | 0.686 | 0.701 | 0.912 | 0.978 |
| P²Net (Yu et al., 2020) | 0.159 | 0.599 | 0.772 | 0.942 | 0.984 |
| Bian *et al.* (Bian et al., 2020) | 0.147 | 0.536 | 0.804 | 0.950 | 0.986 |
| SC-DepthV1 (Bian et al., 2021) | 0.159 | 0.639 | 0.734 | 0.937 | 0.983 |
| PLNet (Jiang et al., 2021) | 0.151 | 0.562 | 0.790 | 0.953 | 0.989 |
| StructDepth (Li et al., 2021) | 0.142 | 0.540 | 0.813 | 0.954 | 0.988 |
| Zhang *et al.* (Zhang et al., 2022b) | 0.177 | 0.634 | 0.733 | 0.936 | - |
| ADPDepth (Song et al., 2023) | 0.165 | 0.592 | 0.753 | 0.934 | 0.981 |
| F²Depth (Guo et al., 2024a) | 0.153 | 0.569 | 0.787 | 0.950 | 0.987 |
| Guo *et al.* (Guo et al., 2024b) | 0.152 | 0.567 | 0.792 | 0.950 | 0.988 |
| **TSF-Depth** | **0.129** | **0.527** | **0.846** | **0.966** | **0.991** |

Table 4: **Generalization resutls on ScanNet** (Dai et al., 2017). All methods are trained on NYUv2.

| Method | Error Metric (↓) | | Accuracy Metric (↑) | | |
|---|---|---|---|---|---|
| | Abs Rel | RMSE | $\delta < 1.25$ | $\delta < 1.25^2$ | $\delta < 1.25^3$ |
| MovingIndoor (Zhou et al., 2019) | 0.212 | 0.483 | 0.650 | 0.905 | 0.976 |
| Monodepth2 (Godard et al., 2019) | 0.200 | 0.458 | 0.672 | 0.922 | 0.981 |
| TrainFlow (Zhao et al., 2020) | 0.179 | 0.415 | 0.726 | 0.927 | 0.980 |
| P²Net (Yu et al., 2020) | 0.175 | 0.420 | 0.740 | 0.932 | 0.982 |
| PLNet (Jiang et al., 2021) | 0.176 | 0.414 | 0.735 | 0.939 | 0.985 |
| IFMNet (Wei et al., 2021) | 0.170 | 0.402 | 0.758 | 0.940 | 0.989 |
| SC-Depthv1 (Bian et al., 2021) | 0.169 | 0.392 | 0.749 | 0.938 | 0.983 |
| StructDepth (Li et al., 2021) | 0.165 | 0.400 | 0.754 | 0.939 | 0.985 |
| **TSF-Depth** | **0.157** | **0.390** | **0.775** | **0.954** | **0.988** |

## 4.4 COMPARISON ON INDOOR SCENE

**NYUv2.** Table 3 presents a performance comparison between our approach and state-of-the-art methods on the NYUv2 dataset (Silberman et al., 2012). For smaller indoor scenes, TSF-Depth significantly outperforms all previous self-supervised methods compared to larger outdoor scenes. This shows that building 3D scene fields is effective and can be easily done in small scenes.

**ScanNet.** We further validate the generalization ability in indoor scenes. All methods are trained on NYUv2 (Silberman et al., 2012) and tested on ScanNet (Dai et al., 2017). The results in Table 4 demonstrates that TSF-Depth has excellent generalization ability for unseen indoor scene.

## 4.5 ABLATION STUDY

To investigate the main contributions and key designs of TSF-Depth, a series of ablation experiments on the KITTI (Geiger et al., 2013) dataset are conducted. The pipeline of the baseline is reported in Fig. 1 (a). In addition, the ablation studies for indoor scene are presented in Appendix F.

**Effects of Multi-scale Scene Priors.** We first analyze the impact of incorporating multi-scale scene prior into model by positional encoding. See Table 5 (a), (b), (d) and (e), training either with single-scale or full-scale scene prior encoding, all significantly improves the depth accuracy over the baseline without it, and the combination it with tri-plane feature scenes at different scales yields an additional improvement. Based on the above analysis, the scene prior we introduce is effective.

**Effects of Multi-scale Tri-plane Feature Fields.** We further analyze the impact of modeling 3D scene field using multi-scale tri-plane feature fields. The results are shown in Table 5 (c) and (e). When building a single-scale tri-plane feature field, the depth accuracy is improved. Better performance is achieved by using multi-scale tri-plane feature field or incorporating scene priors, suggesting that modeling multi-view representation is effective for robust depth estimation.

**Effects of 3D Scene Field on 3D Geometric Representation.** We finally evaluate the effectiveness of 3D scene fields for 3D geometric representation. The results is reported in Table 6. We remove the decoder $\theta_I$ branch and positional encoding branch of TSF-Depth as a baseline and then train it.

Table 5: Ablation results for each component of our method on KITTI (Geiger et al., 2013). $SP^i$: Incorporate scene prior encoding with resolution $\frac{H}{2^i} \times \frac{W}{2^i}$. $TP^i$: Model the 3D scene using tri-plane feature field with resolution $\frac{H}{2^i} \times \frac{W}{2^i}$. $SP^{All}/TP^{All}$: Use all resolution $SP/TP$.

| Exp | Setting | $SP$ | $TP$ | Error Metric ($\downarrow$) | | | | Accuracy Metric ($\uparrow$) | | |
|---|---|---|---|---|---|---|---|---|---|---|
| | | | | Sq Rel | Abs Rel | RMSE | RMSE log | $\delta<1.25$ | $\delta<1.25^2$ | $\delta<1.25^3$ |
| (a) | Baseline | | | 0.746 | 0.102 | 4.464 | 0.176 | 0.897 | 0.965 | 0.983 |
| (b) | $SP^1$ | ✓ | | 0.733 | 0.098 | 4.388 | 0.174 | 0.900 | 0.967 | 0.984 |
| | $SP^2$ | ✓ | | 0.729 | 0.098 | 4.385 | 0.174 | 0.902 | 0.967 | 0.984 |
| | $SP^3$ | ✓ | | 0.722 | 0.098 | 4.386 | 0.175 | 0.901 | 0.967 | 0.984 |
| | $SP^4$ | ✓ | | 0.745 | 0.098 | 4.419 | 0.174 | 0.899 | 0.967 | 0.984 |
| | $SP^5$ | ✓ | | 0.737 | 0.100 | 4.384 | 0.175 | 0.901 | 0.967 | 0.984 |
| | $SP^{All}$ | ✓ | | 0.736 | 0.099 | 4.385 | 0.175 | 0.901 | 0.967 | 0.984 |
| (c) | $TP^1$ | | ✓ | 0.734 | 0.098 | 4.385 | 0.174 | 0.899 | 0.967 | 0.984 |
| | $TP^2$ | | ✓ | 0.738 | 0.098 | 4.387 | 0.174 | 0.901 | 0.967 | 0.984 |
| | $TP^3$ | | ✓ | 0.719 | 0.098 | 4.395 | 0.174 | 0.901 | 0.967 | 0.984 |
| | $TP^4$ | | ✓ | 0.743 | 0.099 | 4.393 | 0.175 | 0.900 | 0.967 | 0.984 |
| | $TP^5$ | | ✓ | 0.760 | 0.099 | 4.425 | 0.175 | 0.898 | 0.967 | 0.984 |
| | $TP^{All}$ | | ✓ | 0.742 | 0.099 | 4.384 | 0.175 | 0.901 | 0.967 | 0.984 |
| (d) | $SP^{All}+TP^1$ | ✓ | ✓ | 0.722 | 0.099 | 4.387 | 0.174 | 0.901 | 0.967 | 0.984 |
| | $SP^{All}+TP^2$ | ✓ | ✓ | 0.719 | 0.098 | 4.385 | 0.174 | 0.902 | 0.967 | 0.984 |
| | $SP^{All}+TP^3$ | ✓ | ✓ | 0.750 | 0.099 | 4.427 | 0.175 | 0.902 | 0.967 | 0.984 |
| | $SP^{All}+TP^4$ | ✓ | ✓ | 0.734 | 0.099 | 4.385 | 0.175 | 0.902 | 0.966 | 0.984 |
| | $SP^{All}+TP^5$ | ✓ | ✓ | 0.748 | 0.100 | 4.406 | 0.175 | 0.902 | 0.967 | 0.984 |
| (e) | $TP^{All}+SP^1$ | ✓ | ✓ | 0.734 | 0.099 | 4.385 | 0.174 | 0.902 | 0.967 | 0.984 |
| | $TP^{All}+SP^2$ | ✓ | ✓ | 0.752 | 0.099 | 4.397 | 0.175 | 0.902 | 0.967 | 0.984 |
| | $TP^{All}+SP^3$ | ✓ | ✓ | 0.740 | 0.099 | 4.435 | 0.175 | 0.901 | 0.966 | 0.984 |
| | $TP^{All}+SP^4$ | ✓ | ✓ | 0.720 | 0.098 | 4.383 | 0.174 | 0.902 | 0.967 | 0.984 |
| | $TP^{All}+SP^5$ | ✓ | ✓ | 0.733 | 0.099 | 4.425 | 0.175 | 0.902 | 0.967 | 0.984 |
| (f) | **TSF-Depth** ($SP^{All}+TP^{All}$) | ✓ | ✓ | **0.692** | **0.096** | **4.335** | **0.173** | **0.903** | **0.967** | **0.984** |

Table 6: Ablation results of the ability of 3D scene fields to represent 3D geometry on KITTI (Geiger et al., 2013). $\theta_E^{PT}$ is an encoder that is frozen after being pre-trained in the baseline.

| Setting | $\theta_E$ | $\theta_E^{PT}$ | $\theta_I$ | $\theta_F$ | Error Metric ($\downarrow$) | | | | Accuracy Metric ($\uparrow$) | | |
|---|---|---|---|---|---|---|---|---|---|---|---|
| | | | | | Sq Rel | Abs Rel | RMSE | RMSE log | $\delta<1.25$ | $\delta<1.25^2$ | $\delta<1.25^3$ |
| Baseline (2D geometry) | ✓ | | | ✓ | 0.746 | 0.102 | 4.464 | 0.176 | 0.897 | 0.965 | 0.983 |
| **TSF-Depth** (2D geometry) | | ✓ | ✓ | ✓ | 0.755 | 0.103 | 4.475 | 0.181 | 0.892 | 0.963 | 0.982 |
| **TSF-Depth** (3D geometry) | ✓ | | ✓ | ✓ | **0.692** | **0.096** | **4.335** | **0.173** | **0.903** | **0.967** | **0.984** |

Thus, the baseline relies on 2D features for depth estimation. Note that we denote the encoder of baseline as $\theta_E^{PT}$. Then, we replace the $\theta_E$ of TSF-Depth with the trained encoder $\theta_E^{PT}$, and then train other modules. In this setup, the Tri-plane feature field is constructed using 2D representation. As for the last row, it uses our complete training. Compared to the baseline and the complete TSF-Depth, we observe that the results of TSF-Depth based on 2D geometry features are worse than both them, due to the 3D-to-2D mapping requiring 3D geometric representation rather than 2D representation. Thus, our TSF-Depth can learn 3D geometric representations for robust depth estimation.

## 5 CONCLUSION

In this paper, we propose a novel self-supervised monocular depth estimation framework based on the **T**hree-dimensional **S**cene **F**ield representation (TSF-Depth). Unlike previous methods typically rely on extracting front-view 2D features, we turn to learning a more powerful 3D representation for robust depth estimation. We design a tri-plane feature field that is reshaped from hybrid features of scene content and scene prior to implicitly model the multi-view representation of the continuous 3D scene. Then, we attentively design a 3D-to-2D mapping strategy to sample 2D features with 3D geometry for depth estimation. Extensive experiments on widely-used outdoor datasets (KITTI and Make3D) and indoor datasets (NYUv2 and ScanNet) show the robustness and generalization ability.

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

## A    OVERVIEW

The appendix document supplements the method details and additional experimental results. In Section B, we discuss the scene priors in detail. In Section C, we supplement the implementation details. In Section D, we present more visualization results on the KITTI (Geiger et al., 2013), Make3D Saxena et al. (2008), NYUv2 Silberman et al. (2012), and ScanNet Dai et al. (2017) datasets. In Section E, we provide more quantitative comparisons with previous state-of-the-art methods at other image resolutions. In Section F, we report the additional ablation study on the indoor scene. In Section G, we present the complexity of the model and the speed of inference. In Section H, we present a challenging sample. In Section I, we present the qualitative results on challenging samples. In Section J, we present the visualization of depth maps and reconstructed point clouds. In Section K, we present the qualitative results of cropped image. In Section L, we discuss the limitations of our method.

## B    DETAILS ABOUT SCENE PRIOR

When humans understand the real-world or infer the depth from the 3D scene including indoor or outdoor scene, they will employ specific prior knowledge about the physical world. For the indoor scene, the floor and celling are located in the lower and the upper parts respectively, and the room is surrounded by flat walls perpendicular to the floor and ceiling. As for the outdoor scene such as driving scene, the road and sky appear in the lower and upper parts respectively, and other objects such as people, vehicles, houses, etc. are connected and located above the road. Besides, objects in the middle area of the image often have greater depth than objects in other areas. Inspired by

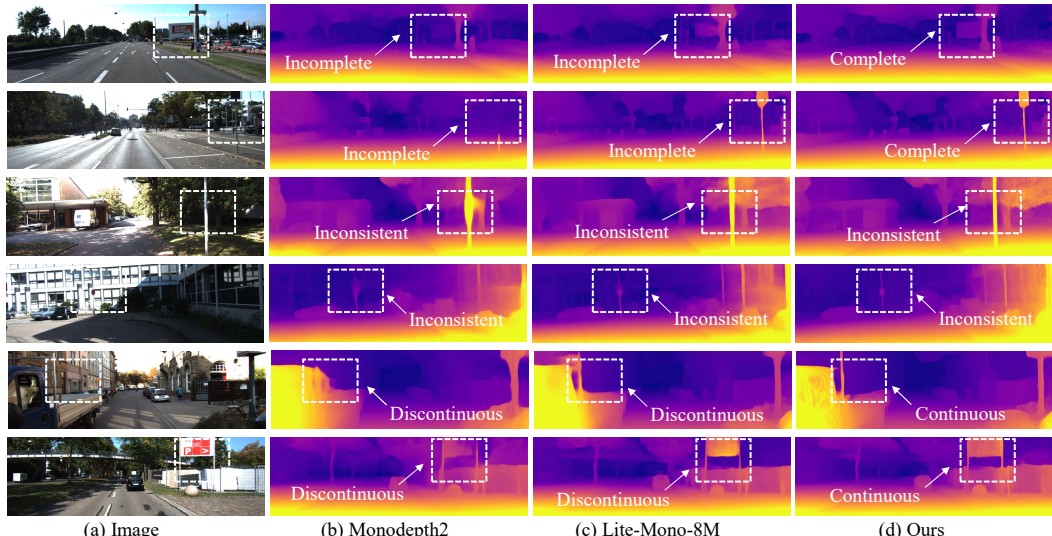

|  |  |  |  |
|---|---|---|---|
| (a) Image | (b) Monodepth2 | (c) Lite-Mono-8M | (d) Ours |

Figure 4: **More qualitative comparison results with a resolution of** $640 \times 192$ **on the KITTI** (Geiger et al., 2013) **dataset** . We highlight challenging areas with white dashed boxes. Compared with the classical method Monodepth2 (Godard et al., 2019) and the latest state-of-the-art method Lite-Mono-8M (Zhang et al., 2023a), our proposed method generates superior visual results.

these observations, we intuitively argue that the rough depth range can be inferred when the scene nature are known and the relative positions of objects in the image are give. To this end, we mine an additional scene prior, *i.e.*, relative spatial position, for depth estimation model to enhance its perception of spatial structure.

## C    MORE IMPLEMENTATION DETAILS

Following Zhou et al. (2021a); He et al. (2022), we use the weights pre-trained on ImageNet (Russakovsky et al., 2015) to initialize the encoders of depth network and pose network. In order to fully verify the effectiveness of the proposed framework, we train and validate outdoor scenes at three resolutions, including $640 \times 192$, $1024 \times 320$ and $1280 \times 384$. And for indoor scenes, we train and validate only at $320 \times 256$ resolution. To improve the training speed, we only output a single-scale depth for the final depth decoder and compute the loss on single-scale depth map. Following existing evaluation rules (Godard et al., 2019; Zhang et al., 2023a), we adopt the same median scaling on the depth results for all methods.

## D    MORE QUALITATIVE RESULTS

For a clear comparison between TSF-Depth and related works, Fig. 4, Fig. 5, and Fig. 14 present more qualitative comparison results on the KITTI and Make3D datasets. In addition, Fig. 6 and Fig. 7 present more qualitative comparison results on the NYUv2 and ScanNet datasets. We highlight challenging areas with white dashed boxes. As we can see from this figure, our proposed TSF-Depth generates more accurate depth maps, particularly in these region with low texture, slender structure, shadow occlusion and reflections.

## E    MORE QUANTITATIVE RESULTS

In order to make more comparisons with previous state-of-the-art methods, we also report the performance comparisons at other resolutions on the KITTI (Geiger et al., 2013) dataset, as shown in Table 7. Note that all methods were trained on both $1280 \times 384$ and $1024 \times 320$ resolutions and then evaluated at the corresponding resolutions. As can be seen, our proposed TSF-Depth significantly outperforms previous self-supervised monocular depth estimation approaches in both resolutions. At a resolution of $1280 \times 384$, almost all errors are reduced by about $3\%$. Specifically, the Sq Rel,

864
865
866
867
868
869
870
871
872
873
874
875
876
877
878
879
880
881
882
883
884
885
886
887
888
889
890
891
892
893
894
895
896
897
898
899
900
901
902

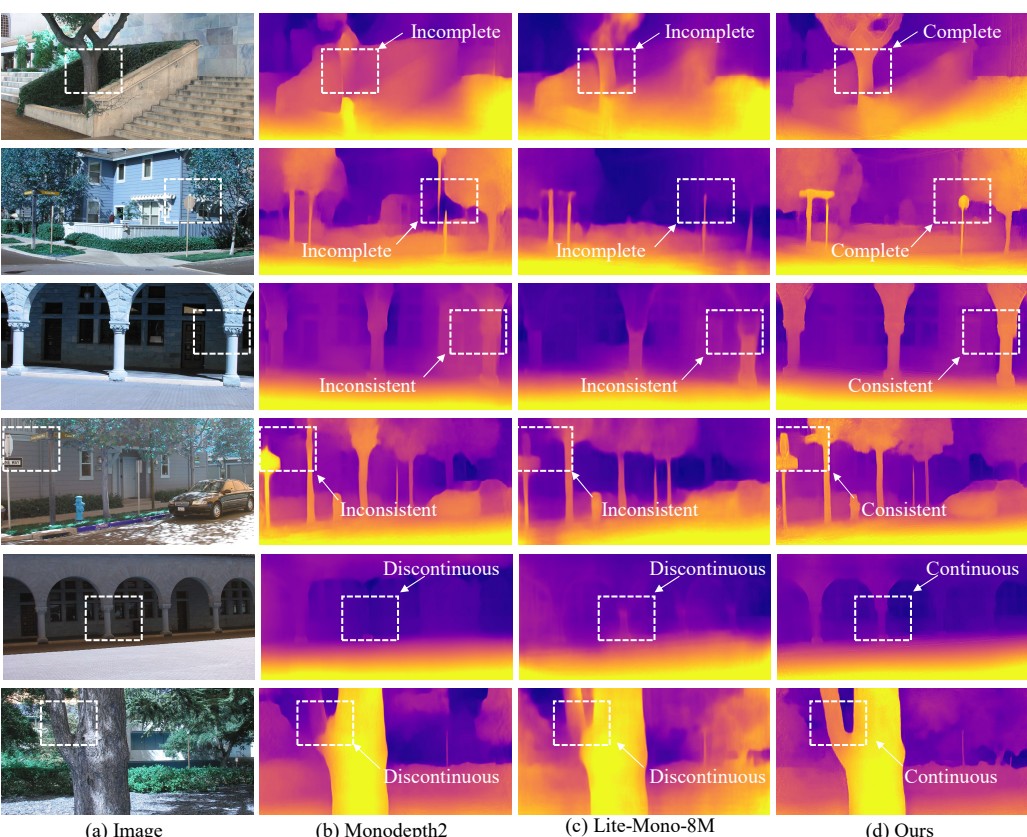

Figure 5: **Qualitative comparison results with a resolution of** $640 \times 192$ **on the Make3D dataset**. We highlight challenging areas. Compared with the classical method Monodepth2 (Godard et al., 2019) and the latest state-of-the-art method Lite-Mono-8M (Zhang et al., 2023a), our proposed method generates superior visual results.

903
904
905
906
907
908
909
910
911
912
913
914
915
916
917

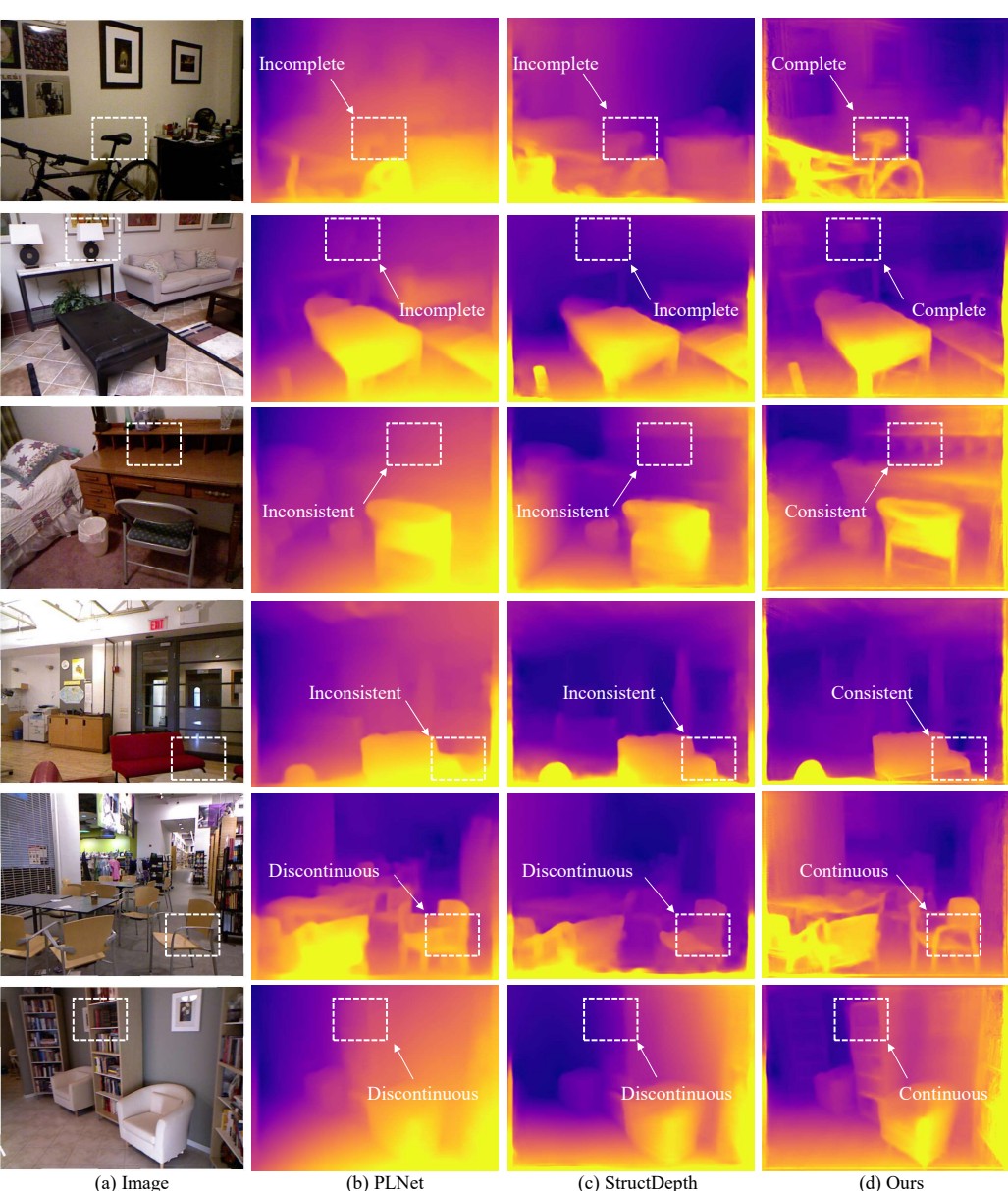

|     |     |     |     |
| --- | --- | --- | --- |
| (a) Image | (b) PLNet | (c) StructDepth | (d) Ours |

Figure 6: **Qualitative comparison results with a resolution of** $640 \times 192$ **on the NYUv2 dataset**. We highlight challenging areas. Compared with the classical method PLNet (Jiang et al., 2021) and the latest state-of-the-art method StructDepth (Li et al., 2021), our proposed method generates superior visual results.

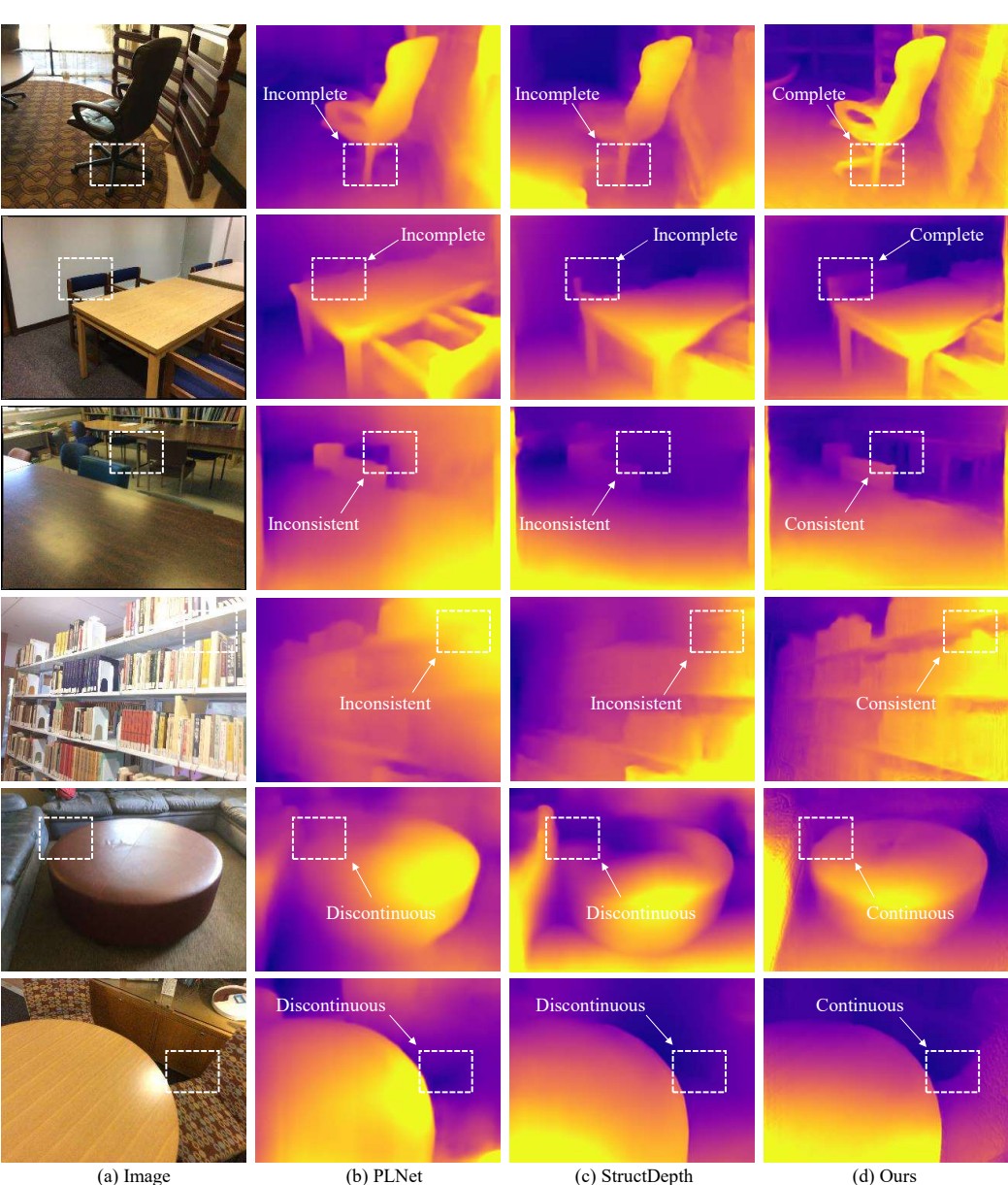

(a) Image       (b) PLNet       (c) StructDepth       (d) Ours

Figure 7: **Qualitative comparison results with a resolution of** $640 \times 192$ **on the ScanNet dataset**. We highlight challenging areas. We highlight challenging areas. Compared with PLNet (Jiang et al., 2021) and the latest state-of-the-art method StructDepth (Li et al., 2021), our proposed method generates superior visual results.

Table 7: **Depth estimation results at other resolutions on the Eigen split of KITTI (Geiger et al., 2013) dataset**. The best results are marked in bold.

| Method | Resolution | Error Metric (↓) | | | | Accuracy Metric (↑) | | |
|---|---|---|---|---|---|---|---|---|
| | | Sq Rel | Abs Rel | RMSE | RMSE log | $\delta<1.25$ | $\delta<1.25^2$ | $\delta<1.25^3$ |
| PackNet (Guizilini et al., 2020) | 1280×384 | 0.758 | 0.104 | 4.384 | 0.182 | 0.895 | 0.964 | 0.982 |
| SGDepth Klingner et al. (2020) | 1280×384 | 0.768 | 0.107 | 4.468 | 0.186 | 0.891 | 0.963 | 0.982 |
| HR-Depth (Lyu et al., 2021) | 1280×384 | 0.727 | 0.104 | 4.410 | 0.179 | 0.894 | 0.966 | 0.984 |
| CADepth (Zhou et al., 2021a) | 1280×384 | 0.715 | 0.102 | 4.312 | 0.176 | 0.900 | 0.968 | 0.984 |
| **TSF-Depth** | 1280×384 | **0.679** | **0.093** | **4.188** | **0.170** | **0.912** | **0.970** | **0.985** |
| Monodepth2 (Godard et al., 2019) | 1024×320 | 0.882 | 0.115 | 4.701 | 0.190 | 0.879 | 0.961 | 0.982 |
| HR-Depth (Lyu et al., 2021) | 1024×320 | 0.755 | 0.106 | 4.472 | 0.181 | 0.892 | 0.966 | **0.984** |
| Lite-Mono (Zhang et al., 2023a) | 1024×320 | 0.746 | 0.102 | 4.444 | 0.179 | 0.896 | 0.965 | 0.983 |
| Zhao *et al.* (Zhao et al., 2024) | 1024×320 | 0.731 | 0.105 | 4.412 | 0.181 | 0.891 | 0.965 | 0.983 |
| **TSF-Depth** | 1024×320 | **0.672** | **0.093** | **4.179** | **0.169** | **0.911** | **0.969** | **0.984** |

Abs Rel, RMSE and RMSE log erros are decreased by $8.8\%$, $5.0\%$, $2.9\%$, $3.4\%$. For the resolution of $1024 \times 320$, almost all errors have a more obvious drop and are reduced by about $5\%$. The Sq Rel, Abs Rel, RMSE and RMSE log erros are decreased by $8.0\%$, $11.4\%$, $5.3\%$, $6.6\%$. Therefore, the quantitative results demonstrate that our designed multi-scale multi-view 3D scene field is more robust to depth estimation at different image resolutions.

## F ADDITIONAL ABLATION STUDY RESULTS

To fully investigate the main contributions and key designs of TSF-Depth, a series of ablation experiments on the NYUv2 (Silberman et al., 2012) dataset are conducted. Compared to outdoor scenes, the challenges in indoor scenes lies in addressing highly diverse environments and near-field clutter with arbitrarily arranged objects. The pipeline for the baseline is reported in Fig. 1 (a), which follows existing self-supervised monocular depth estimation frameworks and employ only front-view 2D geometric feature for depth estimation.

**Effects of Multi-scale Scene Priors.** We first analyze the impact of incorporating multi-scale scene prior into model by positional encoding. As shown in Table 8 (a) and (b), training with either single-scale or full-scale scene prior encoding significantly improves the depth prediction accuracy over the baseline without it. In addition, see Table 8 (d), while combining the multi-scale scene priors with tri-plane feature scenes at different scales does not yield significant improvements, the best depth quality was achieved when combining with multi-scale tri-plane feature fields, as shown in 8 (f). Based on the above analysis, the scene prior we introduce is effective for depth estimation even for monocular challenging indoor scenes.

**Effects of Multi-scale Tri-plane Feature Fields.** We further analyze the impact of modeling 3D scene field using multi-scale tri-plane feature fields. The results are shown in Table 8 (c) and (e). Compared to the baseline, when a single-scale tri-plane feature field is constructed, the depth accuracy is improved. Better performance is achieved by using multi-scale tri-plane or combining scene priors. These ablation results illustrate that our proposed TSF-Depth modeling multi-view 3D scene field representation is effective for robust depth estimation.

## G MODEL COMPLEXITY AND SPEED EVALUATION

Table 9 reports the parameter complexity (#Params), computation complexity (GLOPs), and inference speed on the KITTI (Geiger et al., 2013) dataset. We perform inference at a resolution of $640 \times 1920$, and set the batch size to 16. The models for all comparison methods were inferred on the same platform with NVIDIA RTX 3090 GPU. As can be seen from this table, our model has a similar number of parameters as most existing depth estimation methods, such as Monodepth2-R50 (Godard et al., 2019), DynaDepth (Zhang et al., 2022a) and Zhang et al. (2023b), which allows it to be used on edge devices. Moreover, our model achieve similar computation complexity and inference speed as the existing state-of-the-art model Lite-Mono-8M (Zhang et al., 2023a). Compared

Table 8: **Ablation results for each component of our method on NYUv2** (Silberman et al., 2012). $SP^i$: Incorporate scene prior encoding with resolution $\frac{H}{2^i} \times \frac{W}{2^i}$. $TP^i$: Model the 3D scene using tri-plane feature field with resolution $\frac{H}{2^i} \times \frac{W}{2^i}$. $SP^{All}/TP^{All}$: Use all resolution $SP/TP$.

| Exp | Setting | $SP$ | $TP$ | Error Metric ↓ | | | | Accuracy Metric ↑ | | |
|---|---|---|---|---|---|---|---|---|---|---|
| | | | | Abs Rel | Sq Rel | RMSE | RMSE log | $\delta<1.25$ | $\delta<1.25^2$ | $\delta<1.25^3$ |
| (a) | Baseline | | | 0.147 | 0.123 | 0.574 | 0.189 | 0.795 | 0.956 | 0.990 |
| (b) | $SP^1$ | ✓ | | 0.136 | 0.112 | 0.544 | 0.178 | 0.830 | 0.968 | 0.991 |
| | $SP^2$ | ✓ | | 0.138 | 0.109 | 0.541 | 0.177 | 0.822 | 0.964 | 0.992 |
| | $SP^3$ | ✓ | | 0.138 | 0.116 | 0.559 | 0.177 | 0.823 | 0.965 | 0.991 |
| | $SP^4$ | ✓ | | 0.135 | 0.110 | 0.543 | 0.174 | 0.830 | 0.965 | 0.992 |
| | $SP^5$ | ✓ | | 0.136 | 0.106 | 0.536 | 0.176 | 0.830 | 0.966 | 0.992 |
| | $SP^{All}$ | ✓ | | 0.137 | 0.109 | 0.539 | 0.175 | 0.830 | 0.966 | 0.992 |
| (c) | $TP^1$ | | ✓ | 0.139 | 0.120 | 0.558 | 0.178 | 0.826 | 0.964 | 0.991 |
| | $TP^2$ | | ✓ | 0.143 | 0.113 | 0.551 | 0.183 | 0.812 | 0.959 | 0.991 |
| | $TP^3$ | | ✓ | 0.142 | 0.120 | 0.559 | 0.180 | 0.818 | 0.963 | 0.991 |
| | $TP^4$ | | ✓ | 0.136 | 0.107 | 0.536 | 0.174 | 0.830 | 0.965 | 0.992 |
| | $TP^5$ | | ✓ | 0.138 | 0.115 | 0.548 | 0.177 | 0.829 | 0.963 | 0.991 |
| | $TP^{All}$ | | ✓ | 0.136 | 0.112 | 0.549 | 0.182 | 0.812 | 0.961 | 0.991 |
| (d) | $SP^{All} + TP^1$ | ✓ | ✓ | 0.139 | 0.113 | 0.550 | 0.178 | 0.822 | 0.963 | 0.991 |
| | $SP^{All} + TP^2$ | ✓ | ✓ | 0.137 | 0.112 | 0.549 | 0.176 | 0.826 | 0.965 | 0.991 |
| | $SP^{All} + TP^3$ | ✓ | ✓ | 0.139 | 0.118 | 0.556 | 0.178 | 0.825 | 0.963 | 0.991 |
| | $SP^{All} + TP^4$ | ✓ | ✓ | 0.135 | 0.111 | 0.546 | 0.175 | 0.830 | 0.966 | 0.991 |
| | $SP^{All} + TP^5$ | ✓ | ✓ | 0.135 | 0.109 | 0.536 | 0.176 | 0.829 | 0.964 | 0.991 |
| (e) | $TP^{All} + SP^1$ | ✓ | ✓ | 0.140 | 0.111 | 0.548 | 0.179 | 0.816 | 0.963 | 0.992 |
| | $TP^{All} + SP^2$ | ✓ | ✓ | 0.139 | 0.112 | 0.548 | 0.178 | 0.823 | 0.962 | 0.991 |
| | $TP^{All} + SP^3$ | ✓ | ✓ | 0.141 | 0.115 | 0.554 | 0.180 | 0.816 | 0.961 | 0.991 |
| | $TP^{All} + SP^4$ | ✓ | ✓ | 0.141 | 0.115 | 0.558 | 0.181 | 0.817 | 0.962 | 0.991 |
| | $TP^{All} + SP^5$ | ✓ | ✓ | 0.135 | 0.108 | 0.537 | 0.173 | 0.830 | 0.967 | 0.992 |
| (f) | **TSF-Depth** ($SP^{All} + TP^{All}$) | ✓ | ✓ | **0.134** | **0.103** | **0.530** | **0.172** | **0.831** | **0.967** | **0.992** |

Table 9: **Model complexity and speed evaluation**. We compare parameters (#Params), giga floating-point operations per second (GFLOPS), and inference speed on the KITTI (Geiger et al., 2013) dataset. The input size is $640 \times 192$, and the batch size is 16. All models are inferred on the same platform with NVIDIA RTX 3090 GPU. "-" indicates that the method is not open source code and we cannot make inferences.

| Method | #Params | GFLOPs | Speed | Error Metric (↓) | | | |
|---|---|---|---|---|---|---|---|
| | | | | Sq Rel | Abs Rel | RMSE | RMSE log |
| GeoNet (Yin & Shi, 2018) | 31.6M | - | - | 1.060 | 0.149 | 5.567 | 0.226 |
| Monodepth2-R18 (Godard et al., 2019) | 14.3M | 8.04G | 1.8ms | 0.903 | 0.115 | 4.863 | 0.193 |
| Monodepth2-R50 (Godard et al., 2019) | 32.5M | 16.7G | 2.0ms | 0.831 | 0.110 | 4.642 | 0.187 |
| DynaDepth (Zhang et al., 2022a) | 32.5M | 16.7G | 3.4ms | 0.761 | 0.108 | 4.608 | 0.187 |
| Zhang *et al.* (Zhang et al., 2023b) | 32.6M | - | - | 0.786 | 0.105 | 4.572 | 0.182 |
| Lite-Mono-8M (Zhang et al., 2023a) | 8.70M | 11.2G | 6.3ms | 0.729 | 0.101 | 4.454 | 0.178 |
| Dynamo-Depth (MD2) (Sun et al., 2024) | 14.3M | 8.04G | 1.4ms | 0.864 | 0.120 | 4.850 | 0.195 |
| Dynamo-Depth (Sun et al., 2024) | 8.77M | 11.2G | 4.7ms | 0.758 | 0.112 | 4.505 | 0.183 |
| **Baseline** | 27.2M | 39.9G | 7.7ms | 0.746 | 0.102 | 4.464 | 0.176 |
| **TSF-Depth** | 29.8M | 44.2G | 9.8ms | **0.692** | **0.096** | **4.335** | **0.173** |

to the baseline model, our method scales up to state-of-the-art depth performance with only a few additional parameters. The additional parameters are brought by the initial depth network, however it does not produce any features information to be incorporated into the final depth estimation, only providing sampling points. Therefore, our proposed TSF-Depth is able to be practically applied.

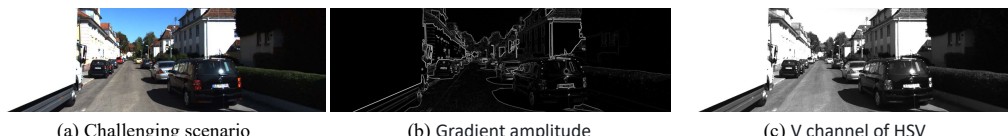

(a) Challenging scenario      (b) Gradient amplitude      (c) V channel of HSV

Figure 8: **A challenging sample**. (a) A scenario with extensive low-texture areas (*e.g.*, roads, buildings, bushes, etc.) and reflective areas (*e.g.*, cars, buildings). (b) The gradient amplitude computed using the Sobel operator to assess the texture distribution. (c) The V channel in HSV color space used to analyze brightness levels.

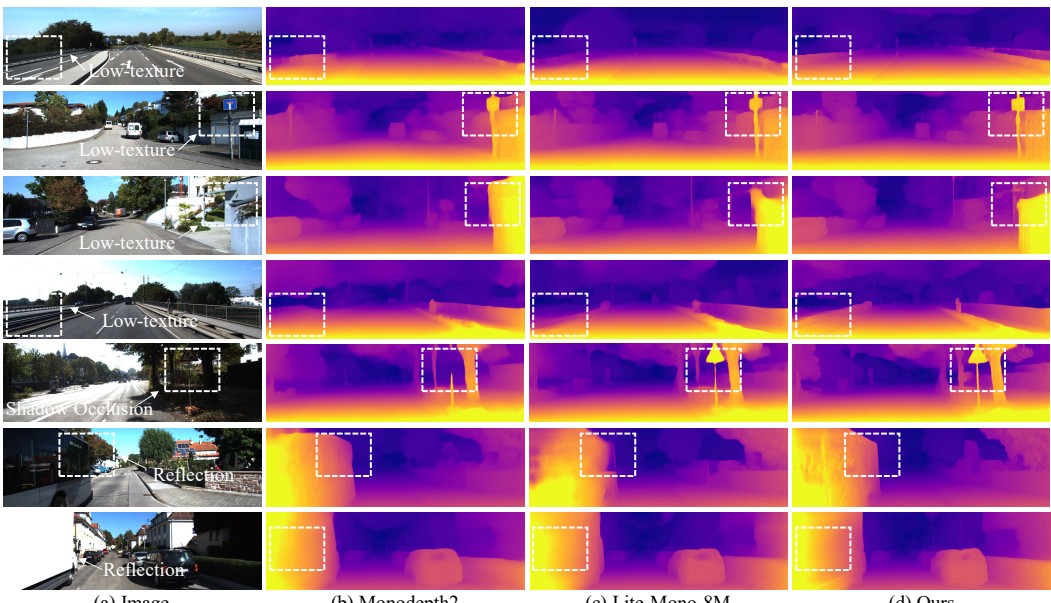

(a) Image      (b) Monodepth2      (c) Lite-Mono-8M      (d) Ours

Figure 9: **Qualitative comparison of challenging test subsets in the KTIT dataset**. We highlight challenging areas with low-texture areas, shadow occlusion, and reflective surfaces. Our method generates robust depth maps in these challenging conditions.

## H  CHALLENGING SAMPLE

In general, low-texture regions exhibit smooth variations with gradient values typically close to zero, while reflective regions often display extremely high luminance values near saturation. We present a challenging sample containing extensive low-texture areas (*e.g.*, roads, buildings, bushes, etc.) and reflective areas (*e.g.*, cars, buildings) in Fig. 8. To analyze these characteristics, we illustrate its gradient magnitude (Fig. 8 (b)) to evaluate the texture level and the V channel in HSV space (Fig. 8 (c)) to examine brightness. The observations align well with common perceptions.

## I  QUALITATIVE RESULTS ON CHALLENGING SAMPLES

We showcase depth estimation results on subsets with low-texture areas, shadow occlusion, and reflective surfaces, as shown in Fig. 9. The highlighted regions (dotted boxes) emphasize the differences among Monodepth2, Lite-Mono-8M, and our method. Our approach demonstrates more accurate and robust depth predictions in these challenging conditions.

## J  VISUALIZATION OF DEPTH MAPS AND RECONSTRUCTED POINT CLOUDS

As illustrated in Fig. 10 (b), we present the visualization results of the depth map at six scales (2st column) alongside the corresponding point clouds without (3nd column) and with (4rd column) color values of the target image. Additionally, we show the reconstructed results (6th and 7th columns) obtained after upsampling the predicted depth map to match the real image resolution, as depicted in Fig. 10 (c). Furthermore, we provide reconstructed results of Fig. 10 from the multiple

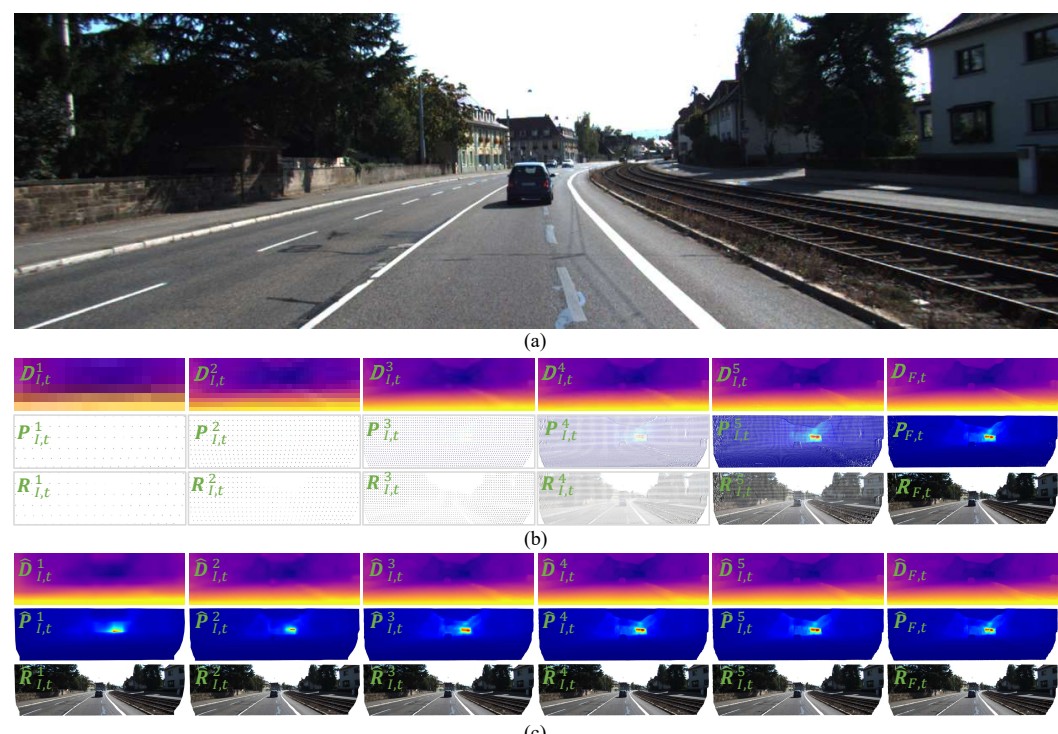

Figure 10: **Qualitative of the depth maps, point clouds without and with color values of the target image**. (a) The target image with a resolution of $375 \times 1242$. (b) Qualitative results at different scales when target image are input at training resolution $192 \times 640$, including predicted depth maps ($\boldsymbol{D}_{I,t}^s \in \mathbb{R}^{192/2^s \times 640/2^s}$ and $\boldsymbol{D}_{F,t}^s \in \mathbb{R}^{192 \times 640}$) (2nd column), projected point clouds ($\boldsymbol{P}_{I,t}^s \in \mathbb{R}^{192/2^s \times 640/2^s \times 3}$, $\boldsymbol{P}_{F,t}^s \in \mathbb{R}^{192 \times 640 \times 3}$, $\boldsymbol{R}_{I,t}^s \in \mathbb{R}^{192/2^s \times 640/2^s \times 3}$ and $\boldsymbol{R}_{F,t}^s \in \mathbb{R}^{192 \times 640 \times 3}$) without (3nd column) and with (4rd column) color values of the target image (c) Qualitative results after the predicted depth is upsampled to the real image resolution.

perspectives in Fig.12. The more qualitative results are reported in Fig. 13. We observed that under the three viewing perspectives, the overall structure of point clouds corresponding to different scales is roughly consistent.

corresponding point clouds without (3nd column) and with (4rd column) color values of the target image

## K    QUALITATIVE RESULTS OF CROPPED IMAGE

As shown in Fig. 11, we present the visualization results of the depth maps corresponding to different cropping ratios. We observe that the Baseline produces discontinuous and inaccurate depth results when the spatial layout of the image changes. In contrast, our method generates consistent results across all three cropping ratios, demonstrating its applicability to challenging cases such as cropped images.

## L    LIMITATION

Although our proposed method has better overall performance in both quantitative and qualitative aspects compared to existing self-supervised methods, there are still limitations. First, insufficient smoothness in glass-related reflective areas, such as the fifth sample in Fig. 4, and the last two samples in Fig. 9. Possible solutions to address this limitation is to incorporate semantic information into the framework, enabling the model to better differentiate reflective regions and enforce smoothness constraints. The second limitation is that predictions may be inaccurate for individual targets that are similar to the overall scene, such as the cabinet in the last sample of Fig. 7. This limitation

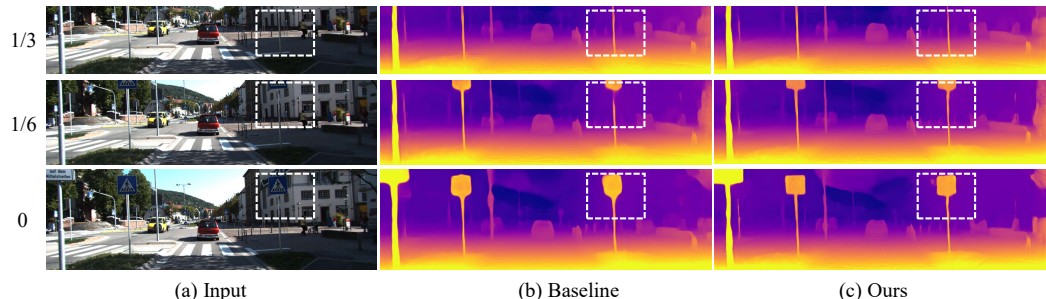

|  | (a) Input | (b) Baseline | (c) Ours |

Figure 11: **Qualitative comparison of cropped images on the KITTI dataset**. We present depth estimation results on images with different crop ratios (1/3, 1/6, and 0). Highlighted regions (dotted boxes) demonstrate that our method achieves more accurate depth predictions compared to the Baseline under varying crop conditions.

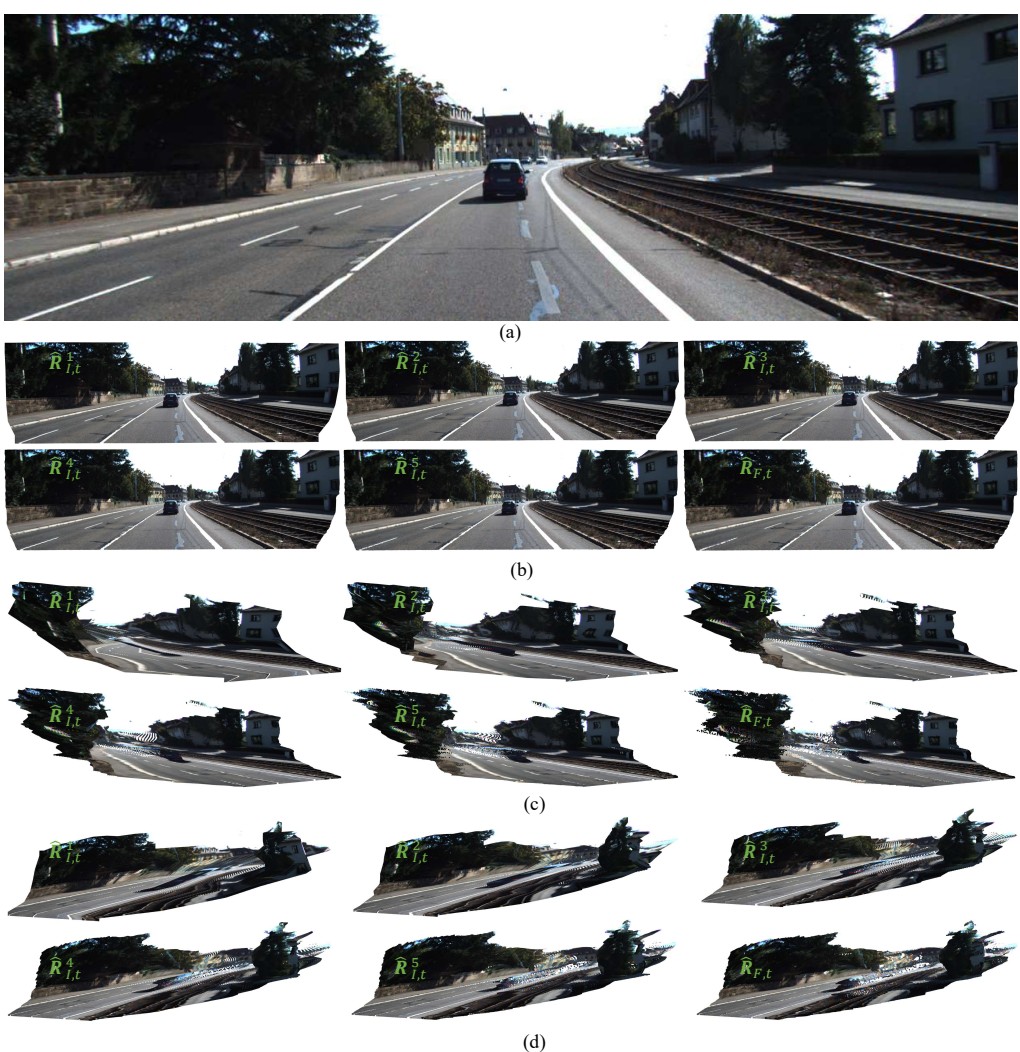

Figure 12: **Qualitative results of reconstructed point clouds from multiple perspectives**. These visualization correspond to Fig. 10.

is likely due to the scarcity of such samples in the training datasets, as these datasets were primarily used for generalization testing. To mitigate this, future work could incorporate semantic information and leverage plane priors to improve predictions in these challenging scenarios.

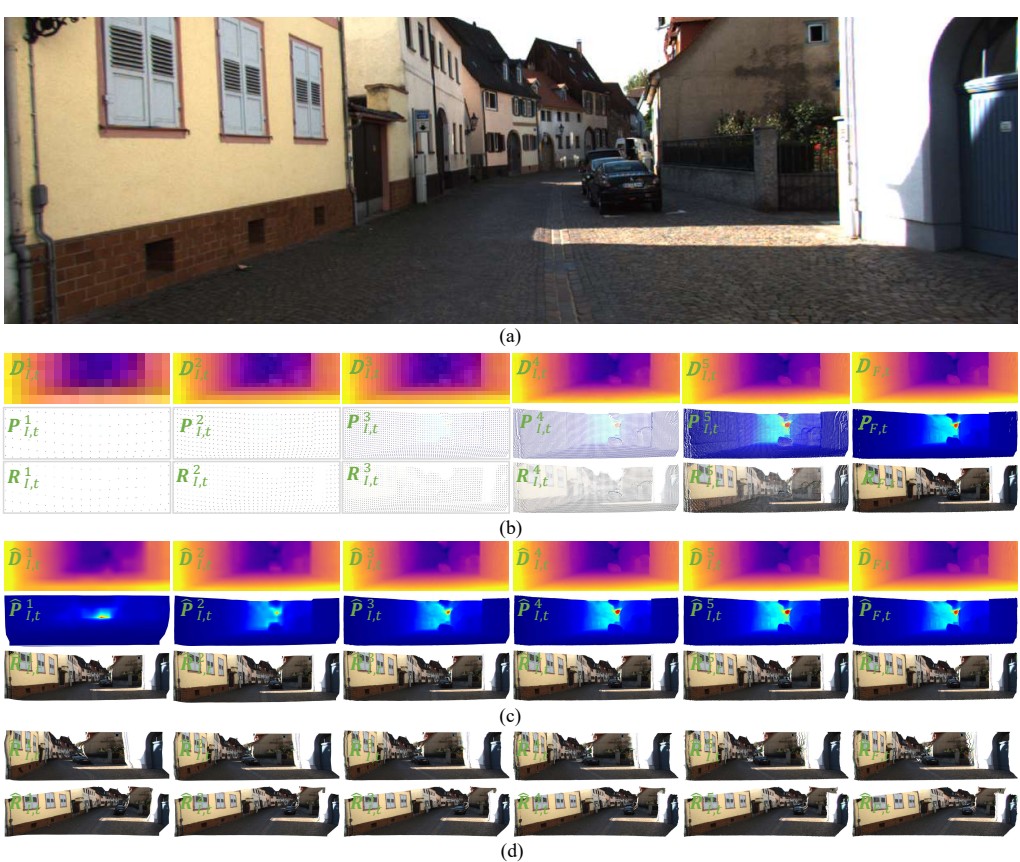

Figure 13: **More qualitative results of the depth maps, point clouds without and with color values of the target image**. (a) The target image with a resolution of $375 \times 1242$. (b) Visualization results at different scales when target image are input at training resolution $192 \times 640$, including predicted depth maps ($\boldsymbol{D}_{I,t}^s \in \mathbb{R}^{192/2^s \times 640/2^s}$ and $\boldsymbol{D}_{F,t}^s \in \mathbb{R}^{192 \times 640}$) (2nd column), projected point clouds ($\boldsymbol{P}_{I,t}^s \in \mathbb{R}^{192/2^s \times 640/2^s \times 3}$, $\boldsymbol{P}_{F,t}^s \in \mathbb{R}^{192 \times 640 \times 3}$, $\boldsymbol{R}_{I,t}^s \in \mathbb{R}^{192/2^s \times 640/2^s \times 3}$ and $\boldsymbol{R}_{F,t}^s \in \mathbb{R}^{192 \times 640 \times 3}$) without (3nd column) and with (4rd column) color values of the target image (c) Qualitative results after the predicted depth is upsampled to the real image resolution. (d) Visualization results of reconstructed point clouds from multiple perspectives.

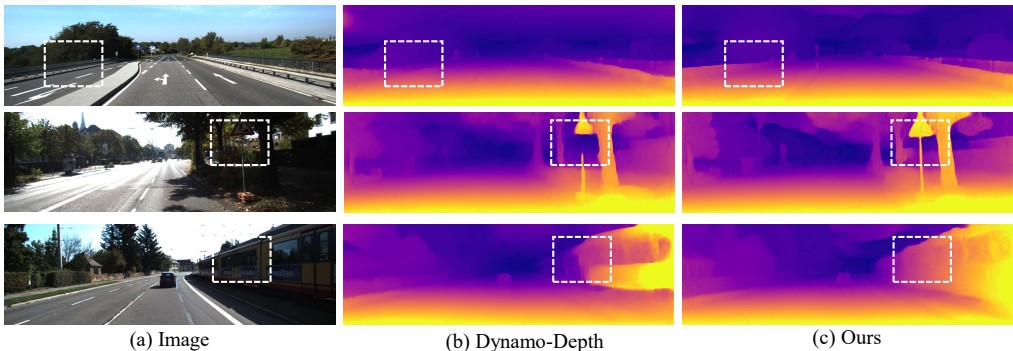

| (a) Image | (b) Dynamo-Depth | (c) Ours |

Figure 14: **Qualitative comparison with recent SOTA work Dynamo-Deth (Sun et al., 2024)**.

