# OpenReview forum: "Beyond 2D Representation: Learning 3D Scene Field for Robust Monocular Depth Estimation"
_ICLR.cc/2025/Conference — ICLR 2025 Conference Withdrawn Submission_

### Official Review · Reviewer_Qiro · 2024-10-30

**Soundness:** 3
**Presentation:** 3
**Contribution:** 2
**Rating:** 5
**Confidence:** 5

**Summary:**

This paper proposes a self-supervised depth estimation network, it designs a 3D intermediate point cloud representation to improve the performance of the network.  And the experiment part is sufficient, the proposed TSF-Depth achieves state-of-the-art performance.

However, I have some slight confusion about this paper in terms of the method design.

**Strengths:**

1. This paper is easy to follow.
2. This paper designs a 3D intermediate point cloud representation for self-supervised depth estimation to improve the performance of the network.
3. The experiment part is sufficient, the proposed TSF-Depth achieves state-of-the-art performance.

**Weaknesses:**

1. The title of this paper should include the keyword "self-supervised".
2. I am certain of the motivation of the paper, but I think there are some unreasonable aspects in the design of the method. I will elaborate on it in detail in the Questions part, and I hope the author can give me the answers.

**Questions:**

I have several questions, If the author can solve my confusion, I will increase the rate.
1. The paper proposes an implicit three-plane representation method, achieving this by decoupling the image coordinates and feature channels of the three dimensions. But, during this "Tri-plane Feature Field" building process, there are no constraints between different plane features. In other words, we can get the 3D representation capability simply by only using different image coordinate encodings?
2. As mentioned in the line 211 "the pixel depth in an image is closely related to its relative spatial position", line 803 "As for the outdoor scene such as driving scene, the road and sky appear in the lower and upper parts respectively, and other objects such as people, vehicles, houses, etc. are connected and located above the road. Besides, objects in the middle area of the image often have greater depth than objects in other areas."
I don't think these priori are always right. In the KITTI dataset, images are taken along the forward direction of the car. However, in many cases, the image may be a side view. Also, to save memory, some scholars may use the top cropping to cut the sky part of image. In this case, is TSF-Depth still applicable for these cropped cases?
3. The paper supervises the generation process of intermediate depth maps through photometric loss.  From another perspective, TSF-Depth is similar to a Coarse-to-Fine network. So, I have a simple question. Can I achieve comparable or even better performance with TSF-Depth by adding a simple refine module on the existing network?
4. I want to know how the network handles sky areas, and I want to see the visualization results about the intermediate depth maps and point cloud projection process.
5. Please state the experimental setting of the ablation experiment more clearly (**Effects of 3D Scene Field on 3D Geometric Representation**). And, as mentioned in line 525, why replace the learnable encoder $\theta_E$ with a pre-trained encoder $\theta_E^{PT}$ without retraining?

---

> ### Author Response · Authors · 2024-11-24
> **Response to Reviewer Qiro**
>
> ***
> >**W1: Add the keyword "self-supervised" to the title.**
>
> >**A:**
> Thanks for your suggestions. We have added the keyword "self-supervised" to the title to improve the clarity of the manuscript.
> ***
> >**Q1: There are no constraints between different plane features. Is it possible to get 3D representation capability just by using different image coordinate encodings?**
>
> >**A1:**
> Although there are no explicit constraints between different plane features, they are implicitly constrained by our 3D-to-2D mapping strategy and end-to-end self-supervised paradigm. Referring to **lines 277-306**, for the generated Tri-plane Feature Field, we project each 3D points onto three orthogonal feature planes to retrieve the corresponding feature via bilinear interpolation, and then aggregate these multi-view geometric feature map via concatenation to predict the depth map. Since the projected geometry is embedded in this mapping process, different plane features can be learned for 3D representation in end-to-end training.
>
> >The image coordinate encoding is treated as a scene priors and is used to enhance the 3D representation by providing spatial cues (see **lines 239-242**). Moreover, the detailed information of the Tri-plane Feature Field is mainly provided by the scene features extracted from the input image.  Therefore, it can not get the 3D representation capability simply by only using different image coordinate encodings.
> ***
> >**Q2: The prior is not always right, and does TSF-Depth still applicable to challenge cases such as side view and cropped image?**
>
> >**A2:**
> In our work, the scene prior is considered as an auxiliary clue to enhance the 3D representation, while the detailed information of the Tri-plane Feature Field is mainly provided by the scene features extracted from the input image.As you correctly pointed out, these priors may not always be accurate. To address this, we employ relative positional encoding instead of absolute image coordinates as the scene prior (see **lines 221-224**). This design shifts the focus of our method from fixed pixel or region positions to their relative relationships, enabling robust depth estimation even in scenarios where the absolute spatial layout undergoes significant changes, such as in side views or cropped images.
>
> > As presented in Table below, we compare depth estimation performance of the Baseline and our proposed TSF-Depth on the KITTI dataset. The evaluation is conducted using top cropping and removing the sky area from the image at different ratios. The crop ratio is reported in the first column, where 0 means no crop. As the cropping ratio increases from 0 to 1/3, progressively removing more of the sky region, the baseline's error metric (Sq Rel) rises from 0.746 to 0.802, representing a 7.51\% increase. In contrast, our proposed TSF-Depth shows a much smaller increase, from 0.692 to 0.704, with only a 1.73\% rise in error. This demonstrates that the priors employed in our method are less sensitive to spatial layout variations. Moreover, the integration of these priors with 3D geometric representation enables our proposed TSF-Depth to achieve robust depth estimation.
>
> >|Crop Ratio|Method|Sq Rel(↓)| Abs Rel(↓)|RMSE(↓)|RMSE log(↓)|δ<1.25(↑)|*δ<1.25²(↑)|δ<1.25³(↑)|
> |:----------------:|--------------|:------------:|:-------------:|:----------:|:--------------:|:------------:|:-------------:|:-------------:|
> | **1/3**        | Baseline| 0.802| 0.104| 4.559| 0.181| 0.896      | 0.964       | 0.982       |
> |                | **TSF-Depth**| **0.704**  | **0.098**   | **4.382**| **0.176**    | **0.900**  | **0.966**   | **0.983**   |
> | **1/6**        | Baseline     | 0.796      | 0.103       | 4.546    | 0.179        | 0.899      | 0.965       | 0.983       |
> |                | **TSF-Depth**| **0.699**  | **0.097**   | **4.370**| **0.175**    | **0.902**  | **0.967**   | **0.984**   |
> | **0**          | Baseline     | 0.746      | 0.102       | 4.464    | 0.176        | 0.897      | 0.965       | 0.983       |
> |                | **TSF-Depth**| **0.692**  | **0.096**   | **4.335**| **0.173**    | **0.903**  | **0.967**   | **0.984**   |
>
> >In addition, we also present the visualization results of the depth maps corresponding to different cropping ratios in **Fig. 11 in Appendix**. We observe that the Baseline produces discontinuous and inaccurate depth results when the spatial layout of the image changes. In contrast, our method generates consistent results across all three cropping ratios, demonstrating its applicability to challenging cases such as cropped images.
>
> ***

---

> > ### Comment · Reviewer_Qiro · 2024-11-26
> > **More Discussion**
> >
> > Thanks to the author's response, but it still does not resolve my confusion.
> >
> > For Q2: To address this, we employ relative positional encoding instead of absolute image coordinates as the scene prior (see lines 221-224).
> >
> > But, in my opinion, the author normalizes the coordinates to [-1,1], so the image coordinates will change between images of different sizes. For example, on an uncropped image (320*1240), if we crop the top 100 pixels, the original pixel at the relative coordinates (0,0) will become (0.72,0), which will affect the final result. How do you deal with this? This is why I doubt the method of this paper.
> >
> > Another question is whether the authors can report on the specific experimental settings in the response.
> >
> > 1. After removing the sky, are the relative coordinates of the image recalculated from the cropped image, or retain the relative coordinates of the original size ?
> > 2. And, what is the baseline method of comparison?
> > 3. Why not remove the sky area entirely (100 pixels, which is a common setting in the field of depth completion), but use a cropping ratio of 0, 1/6, 1/3.

---

> > > ### Author Response · Authors · 2024-11-26
> > > **Adaptability to image size changes**
> > >
> > > >We sincerely thank the reviewer for their follow-up questions and for pointing out these important details. In practical applications, a well-trained depth estimation model might be used in two scenarios: when the resolution of the test image is consistent with the training setting, and when it is inconsistent, such as in cases where the sky portion of the image is cropped to save memory. To evaluate whether our proposed TSF-Depth is applicable to image resolution variations, we consider two distinct experimental setups: (1) training and then testing on datasets with the top region removed at different pixel ratios, and (2) directly testing the performance of the trained depth estimation models, including TSF-Depth, on datasets with the top region removed at different pixel ratios.
> > >
> > > >The first evaluation is presented in the Table of **A2**, where we compare the baseline method with the proposed TSF-Depth framework across a range of cropping ratios (0, 1/6, and 1/3).
> > > The baseline method used in this comparison is a simplified version of the TSF-Depth framework without the tri-plane feature field and positional encoding, which closely resembles existing methods that rely on 2D features for depth estimation, such as the classical Monodepth2 and the SOTA Lite-Mono-8M.
> > >
> > > >By considering multiple cropping ratios, our method is evaluated under a broader range of scenarios, ensuring robustness across different degrees of cropping. This approach is more comprehensive than removing a fixed 100 pixels entirely, as it allows for consistent evaluation across datasets of varying resolutions. For example, with the resolution of KITTI at
> > > $375\times1242$, removing 124 pixels under a 1/3 cropping ratio effectively removes the entire sky region.
> > > The results show that TSF-Depth exhibits a significantly smaller increase in error metrics compared to the baseline when the cropping ratio increases from 0 to 1/3. For instance, baseline increases by 7.51$\%$ (from 0.746 to 0.802), while TSF-Depth increases by only 1.73$\%$ (from 0.692 to 0.704). These results highlight the effectiveness of incorporating 3D geometric representations and relative positional encoding, making TSF-Depth robust to image resolution changes caused by cropping.

---

> ### Author Response · Authors · 2024-11-24
> **Continued Response to Reviewer Qiro**
>
> ***
> >**Q3: Can I achieve comparable or even better performance with TSF-Depth by adding a simple refine module on the existing network.**
>
> >**A3:**
> To evaluate whether comparable or even better performance to TSF-Depth can be achieved by adding a simple refine module on the existing network, we provide an ablation experiment, as shown in Table below:
> |Method| Sq Rel(↓)| Abs Rel(↓)| RMSE(↓)| RMSE log(↓)| δ<1.25(↑) | δ<1.25² (↑) | δ<1.25³ (↑) |#Params |
> |------------|:------:|:------:|:------:|:-------:|:-----:|:--------:|:------:|:-------:|
> |TSF-Depth (Baseline)| 0.746|0.102|4.464|0.176| 0.897|0.965|0.983|27.2M|
> |TSF-Depth (Refine) |0.741|0.101|4.461 |0.176| 0.897|0.965|0.983| 29.6M|
> |**TSF-Depth**|**0.692**|**0.096**|**4.335** |**0.173**| **0.903**|**0.967**|**0.984**| 29.8M|
>
> >For the baseline (row 1), we remove the decoder $\theta_I$ branch and positional encoding branch of TSF-Depth and then train it on the KITTI dataset. Therefore, the pipeline and training strategy of the baseline are almost completely similar to existing self-supervised methods. For the Refine framework (row 2), we remove only positional encoding branch of TSF-Depth, and then use $\theta_F$ as a refine module to optimize the splicing features of the encoder $\theta_E$ and the initial depth. The last row reports the depth results for the full model. It can be seen that the refine strategy has only a slight improvement compared to the baseline. The possible reason is that it still relies on 2D features representation. In contrast, our TSF-Depth has a similar number of parameters as the Refine framework, yet achieves significant improvements due to the 3D geometric representation.
>
> >**Q4: How the network handles sky areas and provide the visualization results of the intermediate depth maps and point cloud.**
>
> >**A4:**
> During the training phase,  most of the sky areas will be masked since we follow Monodepth2 [1] using a masked photometric loss. In this loss function, an auto-masking strategy is designed to generate a per-pixel mask, which is then applied to photometric loss to filter out pixels whose appearance does not change from one frame to the next in the sequence. Therefore, since the sky area has a highly consistent appearance in adjacent frames, it will be removed by mask photometric loss in our framework.
>
>
> >As shown in **Fig. 10 (b) in the Appendix**, we present the visualization results of the depth map at six scales (2nd column) and the corresponding point clouds without (3rd column) and with (4th column) color values of the target image. In addition, we also present the reconstructed results (6th and 7th columns) after the predicted depth map is upsampled to the real image resolution, as shown in **Fig. 10 (c) in the Appendix**. As can be seen in **Fig. 10 (c) in the Appendix**, the reconstructed point clouds at different scales are almost identical, which suggests that our generated depth maps and reconstructed point clouds at different scales are geometrically consistent.
>
> >**Q5: Please state the experimental setting of the ablation experiment more clearly (Effects of 3D Scene Field on 3D Geometric Representation)**
>
> >**A5:**
> Our TSF-Depth aims to achieve robust depth estimation based on the 3D scene representation. In order to evaluate the effectiveness of 3D scene fields for 3D geometric representation, we provide a ablation experiment (*e.g.*, Effects of 3D Scene Field on 3D Geometric Representation) refer to **lines 484-529**. The depth estimation results on the KITTI dataset is reported in **Table 6**, where contains three experimental setting: Baseline (row 1), TSF-Depth based 2D geometry features (row 2), and TSF-Depth based on the 3D geoemtry features (row 3). We remove the decoder $\theta_I$ branch and positional encoding branch of TSF-Depth as a Baseline framework and then train it on the KITTI dataset.  Therefore, the Baseline relies on extracting front-view 2D features for depth estimation. Note that we denote the encoder of Baseline as $\theta_E^{PT}$. Then, to evaluate the effectiveness of $\theta_E$ in learning 3D representation under our end-to-end self-supervised paradigm, we replace the $\theta_E$ of TSF-Depth with the trained encoder $\theta_E^{PT}$, and then train other modules. In this setup, the Tri-plane feature field is constructed using 2D representation. As for the last row in **Table 6**, it uses our complete training. Compared to the Baseline (row 1) and the complete TSF-Depth (row 3), we observe that the results of TSF-Depth based on 2D geometry features are worse than both them, due to the subsequent 3D-to-2D mapping requiring 3D geometric representation rather than 2D representation. Therefore, our TSF-Depth can learn 3D geometric representations during the training phase for robust depth estimation.
> ***
>
> **Reference**
>
> *[1] Godard, Clément, et al. "Digging into self-supervised monocular depth estimation." Proceedings of the IEEE/CVF international conference on computer vision. 2019.*

---

> ### Author Response · Authors · 2024-11-26
> **Adaptability to image size changes**
>
> >The second evaluation focuses on directly testing the trained depth estimation models, including TSF-Depth, on datasets with the top region removed at different pixel ratios.  Following your suggestion, we also report results for cropping 100 pixels, a common setting in the field of depth completion. The results, shown in the following table, compare Monodepth2, Lite-Mono-8M, TSF-Depth, and TSF-Depth*. TSF-Depth uses the relative coordinates of the original image size before cropping, while TSF-Depth* recalculates the relative coordinates based on the cropped image. The "Pixels\&Ratio" column indicates the number of pixels cropped from the top of the image and their corresponding ratio to the total image height. Compared to Monodepth2 and Lite-Mono-8M, both TSF-Depth and TSF-Dept* show minimal performance degradation as the cropping ratio increases, and are always at the leading level. Importantly, in all cases, while the overall performance of TSF-Depth* slightly lags behind TSF-Depth, the performance gap is small. This consistent performance further highlights the robustness of our method in handling spatial layout changes, demonstrating its adaptability and reliability under diverse cropping scenarios.
>
> >| **Pixels&Ratio** | **Method**| **Sq Rel** | **Abs Rel** | **RMSE** | **RMSE log** | **δ < 1.25** | **δ < 1.25²** | **δ < 1.25³** |
> |:-----------------:|--------------|:----------:|:-----------:|:--------:|:------------:|:-------------:|:-------------:|:-------------:|
> | **125 (1/3)**| Monodepth2|1.833| 0.152 | 6.517    | 0.231 | 0.834| 0.937| 0.970|
> || Lite-Mono-8M| 1.410| 0.138| 5.680| 0.214| 0.849 | 0.943 | 0.976|
> || TSF-Depth | 1.245 | 0.130| 5.397| 0.206 | 0.860| 0.950 | 0.977|
> | | TSF-Depth*| 1.232| 0.132| 5.380 | 0.209 | 0.856 | 0.949| 0.977 |
> | **100 (4/15)** | Monodepth2 | 0.973| 0.125 | 5.070 | 0.202| 0.861| 0.955| 0.980|
> | | Lite-Mono-8M| 1.051| 0.122| 4.966| 0.197| 0.869| 0.955| 0.980|
> || TSF-Depth | 0.983| 0.117| 4.881 | 0.193 | 0.875| 0.958| 0.981|
> | | TSF-Depth*| 0.958| 0.118 | 4.833| 0.195| 0.874| 0.958| 0.980|
> | **62 (1/6)**| Monodepth2| 0.867 | 0.117 | 4.867| 0.195| 0.871| 0.958| 0.981|
> | | Lite-Mono-8M| 0.870| 0.113| 4.617| 0.187 | 0.883 | 0.962| 0.983|
> || TSF-Depth| 0.784| 0.105 | 4.488| 0.181| 0.890| 0.965| 0.983|
> | | TSF-Depth*| 0.763 | 0.105| 4.416 | 0.183 | 0.891| 0.965| 0.983 |
> | **0 (0/375)**| Monodepth2 | 0.903| 0.115 | 4.863| 0.193 | 0.877| 0.959| 0.981|
> || Lite-Mono-8M| 0.729| 0.101| 4.454| 0.178| 0.897| 0.965| 0.984|
> |  | TSF-Depth | 0.692| 0.096 | 4.335| 0.173| 0.903 | 0.967 | 0.984|
> || TSF-Depth*| 0.692| 0.096| 4.335| 0.173| 0.903| 0.967| 0.984|
>
> >Based on the above analysis, we can conclude that our proposed TSF-Depth is still applicable to image resolution variations. This robustness stems from the detailed information of the Tri-plane Feature Field is mainly provided by the scene features extracted from the input image. Additionally, the use of relative positional encoding, instead of absolute image coordinates, as the scene prior enhances the 3D representation. This design enables our method to achieve robust depth estimation even in scenarios where the absolute spatial layout undergoes significant changes.

---

> > ### Comment · Reviewer_Qiro · 2024-11-27
> > **More Discussion**
> >
> > Thank you for your reply, but the author seems to be deliberately avoiding my question.
> >
> > 1. First, I theoretically analyzed that the relative coordinates are related to the size of the image, and the author did not refute this point. Therefore, I believe that the design of the relative coordinates does not guarantee the robustness of the results when the image size changes.
> > 2. Why do all the comparison methods significantly improve the RMSE metric after clipping 100 pixels?

---

> > > ### Author Response · Authors · 2024-12-01
> > > **Response to disscussion**
> > >
> > > >Thank you for your patient feedback. We sincerely appreciate your thoughtful insights and emphasis on the potential impact of image size changes on our approach. We agree with the reviewer that the results of our method would be affected by relative coordinates, as the relative coordinates are related to the image size. Although image size changes (*e.g.*, top cropping) introduce intuitive numerical differences in relative coordinates, our TSF-Depth is in fact robust to image size changes due to the combination of relative position encoding and multi-view representation.
> > >
> > > >We would like to illustrate from the following points: **1)** We use multi-view geometric features for depth estimation, which provides greater robustness to image size changes. Top cropping will cause the relative position encoding related to the ordinate to shift, thereby affecting the features of the corresponding part of the Tri-plane Feature Field. However, the features related to the remaining two coordinate dimensions in the Tri-plane Feature Field remain intact, preserving the spatial structure and scene semantics in those dimensions. Thus, even though a shift in the relative coordinates may occur, the impact on depth estimation is mitigated by the robustness of the multi-view features and the preserved structural information in the unaffected dimensions. **2)** Relative coordinates, even with a vertical offset due to cropping, can still provide meaningful relationships between pixels or regions, which are crucial for enhancing depth cues. As shown by Han et al. [1], mining depth cues from consistent horizontal or vertical directions plays a key role in depth estimation tasks. Although the relative coordinates are altered by image size changes, they continue to offer approximate structural and directional information that can guide depth prediction. Specifically, top cropping does not disrupt the horizontal structural information of the image. While cropping introduces a shift in the vertical coordinates, this deviation becomes progressively smaller as you move from the top to the bottom of the image. Consequently, while image size changes may cause shifts in relative coordinates, these shifts remain sufficiently informative for depth estimation, especially when combined with the structural depth cues preserved in the unaffected dimensions.

---

> > > > ### Comment · Reviewer_Qiro · 2024-12-01
> > > > **More Disscussion**
> > > >
> > > > Thank you for your reply.
> > > >
> > > > But, I need to point out the author's mistake.
> > > >
> > > > The author said, "However, the features related to the remaining two coordinate dimensions in the Tri-plane Feature Field remain intact". But, In Eq.3, we can find that, if (u,v) is changed from (0,0) to (0.72,0), all the Tri-plane Features will be impacted.
> > > >
> > > > The authors verified the robustness of their method against cropped images, but this robustness also made me doubt the contribution of the relative coordinates. Even though the author proved that the relative coordinates were valid through ablation experiments, the author failed to solve my doubts and gave me the wrong reply in theoretically.
> > > >
> > > > So I do not increase my score.

---

> > > > > ### Author Response · Authors · 2024-12-02
> > > > >
> > > > > >Thank you for your thoughtful feedback. We sincerely appreciate your attention to the impact of image size changes. In this work, our core motivation is to better address the challenging cases (*e.g.*, low-texture regions, reflective regions, etc.) of depth estimation in real-world scenarios, which are primarily caused by lack of discriminative depth cues.
> > > > > Unlike previous methods that employ only the front-view 2D features, we design a 3D scene field to recover the multi-view representation and then capture sufficient structure- and orientation-aware 3D geometric features from it for robust depth estimation. The 3D scene field is formalized as tri-plane feature field, where we introduce relative position encoding to provide rough structure and use image semantic features to provide details.
> > > > > The standard benchmark setting in the field of depth estimation is to evaluate depth performance on uncropped images.
> > > > > We follow this standard benchmark setting and verify the effectiveness of introducing 3D representation and relative position encoding in depth estimation through quantitative, qualitative, and ablation studies across multiple datasets.
> > > > >
> > > > > >We are grateful for your insight regarding the impact of image size changes on our approach. Here, we would like to further clarify this issue. The image size change issue has been a long-standing challenge in self-supervised monocular depth estimation, and existing approaches often fail in this setting. Although we didn't focus specifically on this challenge, our experiments under different cropping settings demonstrate that our method can more robustly estimate depth for images with size variations, outperforming mainstream depth estimation techniques. The potential reason is that relative positional encoding proposed in our approach, even if it is influenced by the image size, can still provide meaningful relationships between pixels or regions, which are crucial for enhancing depth cues. The acceptable performance gap between using the original size's relative coordinates (e.g., TSF-Depth-OC) and the recalculated relative coordinates (e.g., TSF-Depth-RC) demonstrates the robustness of our TSF-Depth framework to image size changes. However, we are grateful for your valuable comments regarding the issue of image size changes. We have discussed three potential solutions in the previous reply, and would like to continue study this issue indepth in the future.
> > > > >
> > > > >
> > > > > >We hope that our response helps resolve your concerns and sincerely appreciate the opportunity to address your doubts. We believe that the 3D representation proposed in this paper, which is different from the mainstream 2D representation, can make a meaningful contribution to the field and advance monocular depth estimation. To this end, we sincerely hope you can reconsider our work.

---

> > > ### Author Response · Authors · 2024-12-01
> > > **Continued response to disscussion**
> > >
> > > To verify these, we conducted thorough experiments as follows:
> > > **(1)** Retrain and test on cropped images. **(2)** The cropped image is first filled to the original image size, and then the model trained on the original image is directly tested. Note that the reported quantitative results only correspond to the cropped area. **(3)** The cropped image is not filled, and the model trained on the original image is directly used for testing.
> > > The quantitative results of first two validation are reported in **A2** and response to adaptability to image size changes, respectively. These results show that our method is more robust compared to existing methods when training and test image sizes are consistent.The third verification on the KITTI dataset are shown in the following table. Methods marked with $^\bigstar$ indicate performance when existing algorithms are tested directly on original images, with the reported quantitative results corresponding only to the cropped area. Note that methods marked with $^\bigstar$ have consistent quantitative results across different crop ratios because the top 1/3 of the uncropped image was not scanned by the LiDAR [2], and thus there are no depth labels in these areas. Other methods are tested directly on cropped images.  For our proposed method, we report two depth results using the relative coordinates of the original size (denoted as **TSF-Depth-OC**) and using the recalculated relative coordinates based on the cropped image (denoted as **TSF-Depth-RC**).
> > >
> > > >| Pixels&Ratio | Method| Sq Rel(↓) | Abs Rel(↓) | RMSE(↓)  | RMSE log(↓) | δ<1.25(↑) | δ<1.25²(↑) | δ<1.25³(↑) |
> > > |--------------|------------------|--------|---------|-------|----------|---------|---------|---------|
> > > | 125 (1/3)| Monodepth2★| 0.903|0.115| 4.863| 0.193| 0.877| 0.959| 0.981|
> > > || Monodepth2| 2.531  | 0.214| 9.692 | 0.376| 0.665| 0.821| 0.900|
> > > || Lite-Mono-8M★| 0.729| 0.101| 4.454 | 0.178| 0.897| 0.965| 0.983|
> > > || Lite-Mono-8M| 0.884| 0.123| 4.702 | 0.196| 0.866| 0.958| 0.982|
> > > || TSF-Depth★| 0.692| 0.096| 4.335 | 0.173 | 0.903| 0.967| 0.984|
> > > || **TSF-Depth-OC**| 0.779| 0.112| 4.949 | 0.191| 0.864| 0.959| 0.983|
> > > || **TSF-Depth-RC**| 0.810| 0.117| 5.114 | 0.197| 0.864| 0.959| 0.982|
> > > | 100 (4/15)| Monodepth2★| 0.903  | 0.115   | 4.863 | 0.193| 0.877| 0.959| 0.981|
> > > || Monodepth2 | 1.457| 0.196| 6.757 | 0.285| 0.684| 0.901| 0.960|
> > > || Lite-Mono-8M★| 0.729| 0.101| 4.454 | 0.178| 0.897| 0.965| 0.983|
> > > || Lite-Mono-8M| 0.871| 0.119| 4.763 | 0.193| 0.868| 0.959| 0.982|
> > > || TSF-Depth★| 0.692| 0.096| 4.335 | 0.173| 0.903| 0.967   | 0.984|
> > > || **TSF-Depth-OC**| 0.776| 0.111| 4.796 | 0.187| 0.875| 0.962| 0.983|
> > > || **TSF-Depth-RC**| 0.792| 0.110| 4.851 | 0.192| 0.861| 0.961| 0.983|
> > > | 62 (1/6)| Monodepth2★| 0.903  | 0.115| 4.863 | 0.193| 0.877| 0.959| 0.98|
> > > || Monodepth2| 0.972| 0.129| 5.569 | 0.217| 0.834| 0.943| 0.976|
> > > || Lite-Mono-8M★| 0.729| 0.101| 4.454 | 0.178| 0.897| 0.965| 0.983|
> > > | | Lite-Mono-8M| 0.824| 0.110| 4.551 | 0.184| 0.886| 0.963| 0.982|
> > > || TSF-Depth★| 0.692| 0.096| 4.335 | 0.173| 0.903| 0.967| 0.984|
> > > || **TSF-Depth-OC**| 0.708| 0.100| 4.343 | 0.175| 0.899| 0.967| 0.983|
> > > || **TSF-Depth-RC**| 0.705| 0.101| 4.374 | 0.177| 0.897| 0.967|0.984|
> > >
> > > >From this table, we can draw the following conclusions: **(1)** All methods, including our, depth performance degradation when the input resolution between training images and test images is inconsistent. This degradation increases with the crop ratio. However, TSF-Depth-OC and TSF-Depth-RC still maintain better depth performance than existing methods.
> > > The potential reason is that our method models a multi-view representation for depth estimation, rather than relying solely on front-view features. This enables more robust depth estimation even when one dimension of relative coordinates changes, as other dimensions and multi-view scene semantic features help mitigate the impact.**(2)**
> > > TSF-Depth-OC consistently outperforms TSF-Depth-RC in all cropping settings and metrics. As the crop ratio increases, the performance gap between TSF-Depth-OC and TSF-Depth-RC widens, as we use relative coordinates with larger offsets as input. However, the performance gap is always slight. This observation demonstrates that our TSF-Depth is robust to image size changes.

---

> > > ### Author Response · Authors · 2024-12-01
> > > **Continued response to disscussion**
> > >
> > > >To mitigate the impact of relative coordinates  related to image size on our algorithm, we propose the following solution:
> > > **(1)** When the crop ratio is known, we can recover the relative coordinates of the original size based on the current image size, and then adopt the recovered coordinates as input to our model.
> > > **(2)** When the cropping ratio is not known, we can utilize some common crop setting to roughly infer the original size. As you noted, a typical crop of 100 pixels is commonly used in depth completion to remove the sky area. Additionally, in outdoor scenes like KITTI, a crop ratio of 1/3 to 1/4 is typical, as the upper regions of the image usually correspond to the sky, which are not scanned by LiDARs [2]. By combining this prior knowledge with observations of the image content, we can choose an appropriate crop ratio to estimate the original size, thereby reducing the coordinate bias.
> > > **(3)** There may be other methods to improve the robustness to the image size changes such as multi-scale training or more advanced positional encoding, we leave these for our further research.
> > >
> > > >**Why do all the comparison methods significantly improve the RMSE metric after clipping 100 pixels?**
> > > Thank you for pointing out the inconsistencies in the table. Upon rechecking our evaluation, we identified that the cropping by 100 pixels was mistakenly set to 195 pixels in our previous calculations. However, all other results are correct. We have updated the table accordingly.
> > >
> > > >Finally, we would like to express our sincere gratitude for your thoughtful review, and genuinely hope that our responses address your concerns. Please feel free to let us know if additional clarifications are needed. We thank you again for your time and effort.
> > >
> > >
> > > [1] Han, Wencheng, Junbo Yin, and Jianbing Shen. "Self-Supervised Monocular Depth Estimation by Direction-aware Cumulative Convolution Network." Proceedings of the IEEE/CVF International Conference on Computer Vision. 2023.
> > >
> > > [2] Wu, Cho-Ying, and Ulrich Neumann. "Scene completeness-aware lidar depth completion for driving scenario." ICASSP 2021-2021 IEEE International Conference on Acoustics, Speech and Signal Processing (ICASSP). IEEE, 2021.

---

> ### Author Response · Authors · 2024-12-03
> **Kind Reminder: Discussion Deadline Approaching**
>
> Dear Reviewer Qiro,
>
> We would like to sincerely thank you for your thoughtful and valuable feedback on our submission. As the discussion period comes to a close, we would greatly appreciate it if you could let us know whether our responses have addressed your concerns or if there are any additional points you would like us to clarify. If our responses have satisfactorily addressed your concerns, we kindly ask you to consider reflecting this in your score.
>
> Once again, thank you for your time and thoughtful input throughout this process.
>
> Best regards,
>
> Authors of submission 243

---

### Official Review · Reviewer_nMyD · 2024-10-31

**Soundness:** 3
**Presentation:** 3
**Contribution:** 3
**Rating:** 6
**Confidence:** 4

**Summary:**

The authors propose to use the 3d scene field for self-supervised monocular depth estimation. To this end, a tri-plane feature field and a 3D-to-2D mapping strategy are further developed. The proposed method is validated on multiple datasets including KITTI, Make3D, NYU, and ScanNet and outperforms previous methods.

**Strengths:**

The paper is well-written and structured;

The proposed method is novel to the best of my knowledge;

The experiments are thorough and the proposed method outperforms previous state-of-the-art methods.

**Weaknesses:**

No obvious weakness is observed.

**Questions:**

What level of improvement does the proposed method achieve compared to Lite-Mono when using the same encoder? For instance, both use Lite-Mono-8M as the encoder, along with the corresponding computational efficiency, such as the number of parameters and inference time.

---

> ### Author Response · Authors · 2024-11-24
> **Response to Reviewer nMyD**
>
> ***
> >**Q1: How about the performance improvement of our approach when using the same encoder as Lite-Mono?**
>
> >**A1:**
> The following Table presents the depth estimation results on the KITTI dataset when using Lite-Mono-8M as the encoder, comparing the existing method Lite-Mono with the proposed TSF-Depth framework. The table evaluates the performance based on both error metrics (lower is better) and accuracy metrics (higher is better), while also considering model complexity and inference speed.
>
> > The Lite-Mono method achieves a relatively low computational cost with 11.2 GFLOPs and 6.3ms inference time. However, its error metrics, such as Sq Rel (0.832) and RMSE (4.597), indicate suboptimal accuracy. The accuracy metrics ($\delta<1.25$: 0.895) are also slightly lower, suggesting room for improvement in capturing depth-related geometric features. Notably, the Lite-Mono results presented here are reproduced by us based on the official code and experimental settings described in the original paper. However, as discussed in several issues on the official repository (see issues: \#72, \#118, \#124, \#128 \#131, \#141, \#143, etc.), there are differences between the reproduced results and the ones reported in the paper. Our reproduced results are highly consistent with those provided by other researchers in these discussions.
>
> >In contrast, the TSF-Depth framework, when combined with the same Lite-Mono-8M encoder, significantly improves performance. The Sq Rel error is reduced from 0.832 to 0.717, and the RMSE is reduced from 4.597 to 4.355, reflecting better depth prediction accuracy. Moreover, the $\delta<1.25$ accuracy increases from 0.895 to 0.900. These improvements come with a moderate increase in computational cost (16.5 GFLOPs) and a slightly slower inference time (4.4ms).
>
> > In addition, the extra parameters mainly come from $\theta_I$, which predicts point clouds to sample 2D features with 3D geometry from the 3D scene field, rather rather directly providing the features required for depth estimation. Therefore, the depth improvement is due to the 3D representation rather than by increasing network parameters.
>
> >| Method       | Encoder         | #Params | GFLOPs | Speed  | Sq Rel(↓) | Abs Rel(↓) | RMSE(↓)  | RMSE log(↓) | δ<1.25(↑) | δ<1.25² (↑) | δ<1.25³ (↑) |
> |:------------:|:---------------:|:-------:|:------:|:------:|:------:|:-------:|:-----:|:--------:|:------:|:-------:|:------:|
> | Lite-Mono    | Lite-Mono-8M    |  8.7M   | 11.2G  | 6.3ms  | 0.832  |  0.104  | 4.597 |  0.182   | 0.895  |  0.964  |  0.982  |
> | **TSF-Depth**| Lite-Mono-8M    | 11.2M   | 16.5G  | 4.4ms  | **0.717**  |  **0.098**  | **4.355** |  **0.176**   | **0.900**  |  **0.966**  |  **0.984**  |
>
> ***

---

> > ### Comment · Reviewer_nMyD · 2024-11-26
> >
> > Thanks for your response. The authors have addressed most of my concerns.

---

> > > ### Author Response · Authors · 2024-12-03
> > > **Gratitude for Your Feedback**
> > >
> > > Dear Reviewer nMyD,
> > >
> > > We would like to sincerely thank you for your thoughtful and constructive feedback on our work. We genuinely appreciate your acknowledgment of our responses. We greatly appreciate your acknowledgment of our responses, as well as your positive comments regarding the novelty and superior performance of our approach. As the discussion period is nearing its end, we would be grateful for additional questions or points of clarification you might have. If our responses have satisfactorily addressed your concerns, we kindly ask you to consider reflecting this in your score.
> > >
> > > Thanks again for your time and expertise.
> > >
> > > Best regards,
> > >
> > > Authors of submission 243

---

### Official Review · Reviewer_kECm · 2024-10-31

**Soundness:** 3
**Presentation:** 3
**Contribution:** 3
**Rating:** 6
**Confidence:** 4

**Summary:**

The paper proposes a self-supervised monocular depth estimation method that incorporates additional tri-plane features to establish a 3D representation, enhancing robustness in depth estimation which previous self-supervised methods struggle with. Key contributions of this work include:

- Introducing the first 3D scene field for monocular depth estimation using tri-plane features.
- Utilizing 3D-to-2D mapping techniques to project the 3D representation onto 2D image space.
- Conducting diverse experiments across various datasets to validate the approach.

**Strengths:**

S1. Clear Contribution with Tri-plane Features: The paper presents a contribution by incorporating tri-plane features to create 3D representation for self-supervised monocular depth estimation.

S2. Diverse Experiments: The paper shows diverse experiments with multiple datasets and cross validations.

S3. Writing and Presentation: The paper is well-written and easy to understand.

**Weaknesses:**

W1. Clarity of Figure: The paper includes many parameter variables (e.g. Fs, Es, ˜Fs). However, Figure 2 does not contain notations that indicate where each parameter is represented, making it difficult to understand the overall method.

W2.  Limited Qualitative Results: The qualitative results are only provided for KITTI. While the quantitative results show improvements across various datasets, it is unclear whether these improvements are due to the 3D representation or merely a result of an increase in the number of network parameters.

**Questions:**

I would like to start by thanking the authors for their contribution to this field with their submission. Here are some questions:

Q1: This relates to W1. Adding notations to Figure 2 could make it easier to understand and interpret.

Q2: This relates to W2. If there are visualization results for other datasets, it would help in assessing whether the proposed design truly preserves completeness and consistency, as claimed in the paper.

---

> ### Author Response · Authors · 2024-11-24
> **Response to Reviewer kECm**
>
> ***
> >**W1, Q1: Add notations to Fig. 2.**
>
> > **A1:**
> Thanks for your suggestions. We have added all notations including $F^s$, $E^s$, $\bar F^s_{xy}$, $\bar F^s_{xz}$, $\bar F^s_{zy}$, $\tilde F^s_{xy}$, $\tilde F^s_{xz}$, $\tilde F^s_{zy}$ to Fig. 2, which improves the clarity of the manuscript.
> ***
>
> >**W2, Q2: Provide more qualitative results on the remaining three datasets and the reasons for the performance improvements.**
>
> > **A2:**
> Thanks for your suggestion. The visualization results on the remaining three datasets(*i.e.*, Make3D, NYUv2, ScanNet) are presented in **Fig. 5, Fig. 6, and Fig. 7**, respectively, in the Appendix. In addition, we also provide more qualitative results on the KITTI dataset. The visualization results are presented in **Fig. 4 in the Appendix**. We highlight challenging areas with white dashed boxes. From these results we can see that the compared methods produce incomplete depth maps (first two rows), inconsistent depth maps (middle two rows), and discontinuous depth maps (last two rows), while our proposed TSF-Depth generates more accurate depth maps, particularly in these regions with low texture, slender structure, shadow occlusion and reflections.
>
> >To evaluate the depth improvement brought by the 3D representation, we provide an ablation experiment (*i.e.*, **Effects of 3D Scene Field on 3D Geometric Representation**) refer to **lines 484-529 of the manuscript**. The depth estimation results on the KITTI dataset is reported in **Table 6**, where contains three experimental setting: baseline (row 1), TSF-Depth based 2D geometry features (row 2), and TSF-Depth based on the 3D geoemtry features (row 3). We remove the decoder $\theta_I$ branch and positional encoding branch of TSF-Depth as a baseline framework and then train it on the KITTI dataset.  Therefore, the baseline relies on extracting front-view 2D features for depth estimation. Note that we denote the encoder of baseline as $\theta_E^{PT}$. Then, to evaluate the effectiveness of $\theta_E$ in learning 3D representation under our end-to-end self-supervised paradigm, we replace the $\theta_E$ of TSF-Depth with the trained encoder $\theta_E^{PT}$, and then train other modules. In this setup, TSF-Depth based 2D geometry features has same network parameters as TSF-Depth based on the 3D geoemtry features, and the Tri-plane feature field is constructed using 2D representation. As for the last row in Table 6, it uses our complete training. Compared to the baseline (row 1) and the complete TSF-Depth (row 3), we observe that the results of TSF-Depth based on 2D geometry features are worse than both them, due to the subsequent 3D-to-2D mapping requiring 3D geometric representation rather than 2D representation. Therefore, the depth improvement comes from the 3D representation.
>
> >To evaluate the depth improvement is not due to an increase in the number of network parameters, we report the parameter complexity of the baseline, TSF-Depth, and some other compared methods in **Table 9 of the manuscript**. As can be seen from this table, TSF-Depth achieves the best performance with a similar number of parameters to the baseline and most existing depth estimation methods. Therefore, the depth improvement is due to the 3D representation rather than by increasing network parameters.
> ***

---

> > ### Comment · Reviewer_kECm · 2024-11-25
> >
> > Thank you for the detailed response. This clarifies most of my questions.

---

> > > ### Author Response · Authors · 2024-12-03
> > > **Gratitude for Your Feedback**
> > >
> > > Dear Reviewer kECm,
> > >
> > > Thank you for your thoughtful and constructive feedback on our work. We sincerely appreciate your acknowledgment of our responses. We greatly appreciate your positive comments regarding the clarity of our contribution and the quality of writing. As the discussion period nears its end, we would greatly appreciate any additional questions or points of clarification. If our responses have satisfactorily addressed your concerns, we kindly ask you to consider reflecting this in your score.
> > >
> > > Thanks again for your time and expertise.
> > >
> > > Best regards,
> > >
> > > Authors of submission 243

---

### Official Review · Reviewer_ypGr · 2024-11-01

**Soundness:** 2
**Presentation:** 3
**Contribution:** 2
**Rating:** 5
**Confidence:** 5

**Summary:**

This paper argues that existing monocular depth estimation methods primarily focus on extracting front-view 2D features while neglecting 3D representations, which hinders their performance in various challenging real-world scenarios, including reflections, shadow occlusions, and low-texture regions. To address these limitations, the authors propose a 3D scene field representation within a self-supervised monocular depth estimation pipeline. They have conducted several ablation studies to validate the effectiveness of the proposed modules. Overall, the manuscript is well-structured and easy to follow.

**Strengths:**

The paper proposes multi-scale scene feature and tri-plane feature fields to model a multi-view representation. Based on the ablation study, the proposed method can improve depth accuracy. Although this idea is firstly applied in monocular depth estimation, it has been widely applied in single-image 3D generation.

**Weaknesses:**

1) The paper posits that the incorporation of 3D features enhances robustness and addresses complex real-world scenarios effectively. However, it primarily relies on an ablation study to assess accuracy improvements. While various techniques exist for optimizing model training to enhance performance, mere improvements in accuracy do not necessarily validate the robustness of the proposed features, particularly regarding their argued low-texture regions and reflective surfaces. It is advisable for the authors to include more relevant evaluations that present greater challenges.


2) In Ln267-Ln269, the paper asserts that the predicted multi-scale depths are affine-invariant, leading to the proposal of reconstructing multi-scale point clouds to address distortions induced by affine transformations. However, it is important to note that while an unknown scale does not result in geometric distortions, a shift does. Given that this shift is unrecoverable, the projection process cannot yield a geometrically coherent point cloud. Consequently, I question the validity of the proposed 3D scene field. If the proposed method can mitigates the shift issues, the paper should provide a mathematical analysis of how their method handles this.

3) Because of the shift issues, I suggest that the authors include visualizations of their reconstructed point clouds at different scales to demonstrate their geometric coherence.

4) Where the performance advantage comes from is a bit confusing. The baseline accuracy in Table 5 is already very high, and is almost achieved SOTA compared with methods in Table 1. The question arises as to whether the SOTA performance of TF-DEPTH on benchmarks will come from baseline? In order to make the results more convincing, I suggest to compare the baseline in Table 2,3 and 4.

**Questions:**

1. The paper lacks more convincing experiments to support the method. Although the accuracy is improved, it cannot support the method can solve more complicated real-world scenarios, such as reflections, shadow occlusions, and low-texture regions. Therefore, I suggest the paper design a test set, which includes a higher proportion of challenging cases  (reflections, low-texture regions, etc.). In presentation, the paper could illustrate more qualitative comparisons.

2. As pointed in W2, i.e. affine-invariant distortions caused by the shift, the paper should consider how to model the shift in the unprojection process or provide a mathmatical analysis about how to mitigate the shift issues.

---

> ### Author Response · Authors · 2024-11-24
> **Response to Reviewer ypGr**
>
> ***
> > **W1, Q1: Design a challenging test set to assess the robustness of the proposed approach and provide more qualitative comparisons.**
>
> > **A1:** Thanks for your suggestions. In general, low-texture regions exhibit smooth variations with gradient values typically close to zero, while reflective regions often display extremely high luminance values near saturation. In the Appendix, we present a challenging sample (**Fig. 8 (a)**) containing extensive low-texture areas (*e.g.*, roads, buildings, bushes, etc.) and reflective areas (*e.g.*, cars, buildings). To analyze these characteristics, we also present its gradient magnitude (**Fig. 8 (b)**) and the V channel in HSV space (**Fig. 8 (c)**). We observe that the gradient the gradient of low textures is close to 0, while the brightness of reflective areas approaches saturation, which aligns well with common perceptions.
>
> > Leveraging these properties, we can effectively quantify the low-texture and reflection levels of individual pixels by calculating their gradient magnitudes and brightness values, respectively. Furthermore, the overall difficulty of an image can be assessed by integrating its gradient and brightness scores. To achieve this, we utilize the classical Sobel operator to compute gradient scores and the V channel of HSV space to compute brightness scores, subsequently aggregating them as the basis for selecting the challenging test set.
>
> > Specifically, given an image, its challenging score can be defined as $F(I) = T_{\text{low-texture}} + R_{\text{reflection}}$, where $T_{\text{low-texture}} = \frac{\text{Count}(\nabla I < 2)}{\text{Total pixels}}$, and  $R_{\text{reflection}} = \frac{\text{Count}(\nabla V > 0.6)}{\text{Total pixels}}$. The $\nabla I$ is the gradient amplitude of $I$ calculated based on the Sobel operator, and $\nabla V$ is the $V$ channel in the HSV space.
> To evaluate the performance of TSF-Depth under challenging test sets of different difficulty, we compute the maximum score $F_{max}$ and minimum score $F_{min}$ of the entire test set and then select challenging test sets  $C =${$I, F(I) > \{F_{min}} + \alpha \cdot (F_{max} -  F_{min})$} based on the threshold $\alpha$. We set six thresholds and compared with the classic Monodepth2 and SOTA Lite-Mono-8M on the KITTI dataset, as shown in Table below:
> |α    | Selected/Total | Method| Sq Rel (↓)| Abs Rel (↓)| RMSE (↓) | RMSE log (↓) | δ < 1.25 (↑) | δ < 1.25² (↑) | δ < 1.25³ (↑) |
> |:------:|:----------------:|----------------|:--------:|:---------:|:-------:|:----------:|:----------:|:-----------:|:-----------:|
> | | | Monodepth2|1.554|0.144|6.212 |0.211|0.837|0.946 |0.978|
> |0.9|14/697| Lite-Mono-8M  |1.523|0.132| 5.852 | 0.201| 0.863| 0.954| 0.979|
> | | |**TSF-Depth**| **1.193**| **0.122**| **5.441** |**0.188**|**0.865**|**0.961**|**0.983**|
> | | | Monodepth2|0.997|0.120| 5.111|0.191|0.891|0.959|0.983|
> |0.7   |86/697| Lite-Mono-8M  |0.848  |0.107| 4.679|0.175| 0.894| 0.965| 0.984|
> | |   |**TSF-Depth**|**0.741**| **0.100**| **4.170**| **0.168**|**0.902**|**0.970** |**0.986**|
> |   |  | Monodepth2| 0.906|0.115| 4.963|0.194|0.896|0.959|0.981|
> |0.5|291/697|Lite-Mono-8M|0.763| 0.103 | 4.564 | 0.180 | 0.894| 0.965| 0.984|
> |  |  |**TSF-Depth**|**0.711**|**0.099**|**4.419** |**0.175**|**0.902**|**0.967**|**0.984** |
> |   | | Monodepth2|0.916|0.115|4.909 |0.194|0.877|0.959|0.981|
> |0.3|627/69|Lite-Mono-8M|0.733|0.102|4.507|0.180| 0.889|0.965|0.984|
> | |  |**TSF-Depth**|**0.702**|**0.097**|**4.382** |**0.174**|**0.902**|**0.967**|**0.984** |
> |   |   | Monodepth2|0.906|0.114|4.435|0.193|0.877|0.959|0.981|
> |0.1| 693/697|Lite-Mono-8M| 0.733|0.101|4.400 |0.179|0.889|0.965|0.984|
> |  |  |**TSF-Depth**|**0.692**|**0.096**|**4.335** |**0.175**|**0.903**|**0.967**|**0.984**|
> | |  |Monodepth2|0.903|0.115|4.863|0.193|0.877|0.959| 0.981|
> | 0 |697/697|Lite-Mono-8M|0.733|0.101|4.400|0.179|0.889|0.965|0.984|
> | |  |**TSF-Depth**|**0.692**|**0.096**|**4.335**|**0.175**|**0.903**|**0.967**|**0.984**|
>
> >As the threshold increases, all methods show different performance degradation due to the increasingly challenging test set, while our TSF achieves the best performance on all test sets. This observation highlights the robustness of our approach in dealing with difficult scenarios.
>
> >We provide more qualitative results on a subset of tests with threshold of 0.9, as shown in **Fig. 9 in the Appendix**. As can be seen that these samples contain many low-texture or reflection regions, and slender structure, and existing methods are unable to estimate accurate depth results, whereas our method generates robust depth maps. In addition, we also provide more qualitative results on the Make3D, NYUv2 and ScanNet datasets, as shown in **Fig. 5, Fig. 6, and  Fig. 7 in the Appendix**. As we can see from these visualization results, our proposed TSF-Depth generates more accurate depth maps, particularly in these region with low texture, slender structure, shadow occlusion and reflections.
> ***

---

> > ### Author Response · Authors · 2024-11-24
> > **Continued response to Reviewer ypGr**
> >
> > ***
> > >**W2,Q2: Provide a mathematical analysis on how to mitigate the shift issues.**
> >
> > >**A2:** The un-projection process from 2D coordinates and depth to 3D points is defined as:
> > $\{x = \frac{u - u_0}{f} d,
> > y = \frac{v - v_0}{f} d,
> > z = d\}$, where $(u_0, v_0)$ are the camera optical center, $f$ is the focal length, and $d$ is the depth. As you point out, an unknown scale (*i.e.*, $f$) does not result in geometric distortions as it scales the $x$ and $y$ coordinates uniformly and does not affect $z$. Instead, the depth shift $\Delta d$ affects the $x$, $y$ and $z$ coordinates non-uniformly, which results in shape distortions. In self-supervised paradigm, the shift problem is typically caused by the network's insufficient understanding of the global scene or poor local consistency, resulting in global or local shifts in the predicted depth map. Therefore, we introduce multi-scale depth prediction and multi-scale consistency loss (**Eq. (13) in the manuscript**) to ensures that depth maps at different scales are consistent to reduce shifts. Let the predicted depth at scale $s$ be $\hat{D}_s(x, y) = \alpha D(x, y) + \beta_s$, where $\alpha$ is a global scaling factor, $\beta_s$ is a scale-dependent shift, and $D$ is the depth map. The multi-scale consistency loss ensures that $\beta_s$ approaches zero across all scales during training, effectively aligning the multi-scale depths to a unified, shift-minimized representation. In the un-projection process, this alignment ensures that depth distortions arising from shifts are minimized, producing geometrically coherent point clouds.
> > ***
> >
> > ***
> > >**W3: Provide visualization of reconstructed point clouds at different scales.**
> >
> > >**A3:**
> > As shown in **Fig. 10 (b) in the manuscript**, we present the visualization results of the predicted depth map at six scales (2nd column) and the corresponding point clouds without (3rd column) and with (4th column) color values of the target image. In addition, we also present the reconstructed results (6th and 7th columns) after the predicted depth map is upsampled to the real image resolution, as shown in **Fig. 10 (c)**. As can be seen in **Fig. 10 (c)**, the reconstructed point clouds at different scales are almost identical, which suggests that our generated depth maps and reconstructed point clouds at different scales are geometrically consistent.
> > ***
> >
> > ***
> > >**W4: Compare the Baseline in Table 2, 3 and 4.**
> >
> > >**A4:**
> > These three Tables correspond to the different dataset: the outdoor dataset Make3D (i.e., Table 2 in manuscript) and the indoor datasets: NYUv2 (i.e., Table 3 in manuscript) and ScanNet (i.e., Table 4 in manuscript).
> > The depth estimation results of the Baseline on these datasets are reported in following  two Tables:
> > | Dataset | Method                              | Sq Rel (↓) | Abs Rel (↓) | RMSE (↓)  | RMSE log (↓) |
> > |:---------:|-------------------------------------|:--------:|:---------:|:-------:|:----------:|
> > |   | HR-Depth       | 3.208  | 0.315   | 7.024 | 0.159    |
> > |         | DIFFNet        | 3.313  | 0.309   | 7.008 | 0.155    |
> > |         | Lite-Mono     | 3.060  | 0.305   | 6.981 | 0.158    |
> > |   Make3D      | Lite-Mono-8M   | 3.144  | 0.309   | 7.016 | 0.158    |
> > |         | Zhao et al.     | 3.200  | 0.316   | 7.095 | 0.158    |
> > |         | **Baseline**                       | 3.206  | 0.311   | 7.012 | 0.157    |
> > |         | **TSF-Depth**                      | **2.925** | **0.292** | **6.744** | **0.150** |
> >
> > >| Dataset  | Method                          | Abs Rel (↓) | RMSE (↓)  | δ<1.25 (↑) | δ<1.25² (↑) | δ<1.25³ (↑) |
> > |:----------:|---------------------------------|:---------:|:-------:|:--------:|:---------:|:---------:|
> > | NYUv2    | SC-DepthV1 (Bian et al., 2021) | 0.159   | 0.639 | 0.734  | 0.937   | 0.983   |
> > |          | StructDepth   | 0.142   | 0.540 | 0.813  | 0.954   | 0.988   |
> > |          | **Baseline**                   | 0.147   | 0.574 | 0.795  | 0.956   | 0.990   |
> > |          | **TSF-Depth**                  | **0.129** | **0.527** | **0.846** | **0.966** | **0.991** |
> > | ScanNet  | SC-DepthV1  | 0.169   | 0.392 | 0.749  | 0.938   | 0.983   |
> > |          | StructDepth   | 0.165   | 0.400 | 0.754  | 0.939   | 0.985   |
> > |          | **Baseline**                   | 0.168   | 0.413 | 0.752  | 0.941   | 0.986   |
> > |          | **TSF-Depth**                  | **0.157** | **0.390** | **0.775** | **0.954** | **0.988** |
> >
> > >Fron these results, we find that the Baseline framework cannot directly achieve SOTA performance, while the proposed TSF-Depth achieves SOTA performance. Therefore, the SOTA performance of TSF-Depth in the benchmarks is not come from Baseline.
> > ***

---

> > > ### Comment · Reviewer_ypGr · 2024-11-25
> > > **Discussion on shift**
> > >
> > > In Eq. 5, the paper does not explicitly model the shift in the model optimization. If the shift is ignored in modeling and loss parts, how could the multi-scale consistency loss ensure the shift is minimized to zero?
> > > Shift is an important problem in affine-invariant depth. Many recent papers have tried to solve it. They all explicitly model the shift in the monocular depth learning, i.e. designing another module for it or decoupling it in the loss. However, this paper does not present any design in the training.
> > > If I miss some details. please point out it. However, I believe the provided analysis cannot solve my concern. The shift may cause serious geometry distortions.

---

> > > ### Comment · Reviewer_ypGr · 2024-11-25
> > > **Discussion on Point Cloud Visualization**
> > >
> > > Could authors provide more multi-view visualization of the reconstructed point clouds? It will be much easier to evaluate the geometry consistency. Fig.10 still presents the point cloud in the camera view, which is hard to recognize.

---

> > ### Comment · Reviewer_ypGr · 2024-11-25
> > **discussion**
> >
> > Thanks for the authors' response. I still have concerns about the method's robustness and generalization on reflective regions.
> >
> > In Fig. 9. the method does not work well on reflective regions, such as the window of the bus and the car in the last two cases.  In the fifth example of  Fig. 4, the method also fails on the window regions.
> >
> > In Fig. 7, the paper also compares other methods on Scannet. However, the method fails to predict the depth of the cabinet.
> >
> > In Fig.5 - Fig. 7, the paper indeed performs better than other methods. However, from the visual results, I believe the predicted depth is not promising, and it seems that the model does not converge well. The compared methods are not the SOTA methods.
> >
> > I believe that the visual results are the best examples of the method. However, the presented examples cannot convince me that the method is robust on reflective regions and generalizes to other datasets.

---

> ### Author Response · Authors · 2024-11-25
> **Response to reflective regions**
>
> >Thank you for your detailed feedback and thoughtful comments. We appreciate the opportunity to address your concerns about the robustness and generalization of our method, particularly regarding reflective regions, model convergence, and generalization across datasets.
>
> >Reflective regions, such as bus and car windows in Fig. 9 or the window in Fig. 4, indeed pose significant challenges for monocular depth estimation methods due to their inherent ambiguity and the mismatch between visual appearance and scene geometry. These areas create depth ambiguities that make accurate predictions difficult for all existing approaches, including ours. While our method is not completely flawless in these regions, it is important to highlight that in most cases our approach produces depth maps with relatively complete structure in these regions, which is vital for overall scene understanding. Additionally, It is also important to emphasize that our approach is not limited to addressing challenges in reflective regions.
> Instead, we take a holistic view of monocular depth estimation and address a variety of key challenges that are equally important, including low-texture regions, thin structures, and shadow occlusions. As shown in **Figs. 4–7 and Fig. 9** of the manuscript, our method can smoothly predict low-texture areas, better preserves the geometry of fine details, and more effectively handles shadow occlusion regions. Moreover, our method also has significant advantages over existing methods in challenging scenarios (see table in **A1**). By considering these multiple aspects simultaneously, our method demonstrates superior overall performance across diverse scenarios.
>
> > In qualitative comparison, we selected influential methods in self-supervised monocular depth estimation, including Monodepth2, Lite-Mono-8M, PLNet, and StructDepth. Monodepth2 is a widely recognized baseline in the field of monocular depth estimation and has been extensively used as a benchmark in many recent works. Lite-Mono-8M represents a SOTA method for outdoor scenarios, specifically optimized for real-world driving datasets such as KITTI. PLNet and StructDepth focus on indoor scene depth estimation and, while not the latest works, are highly representative due to their open-source implementations and consistently competitive performance in indoor datasets like NYUv2 and ScanNet. These methods cover a broad spectrum of scenarios, from outdoor driving datasets to complex indoor environments, making them suitable benchmarks for evaluating both robustness and generalization.
>
> >Although the  quantitative results in Table 1, Table 2, Table 3, and Table 4 of the manuscript strongly support that our model has converged well, we will carefully revisit our training pipeline to ensure that no potential improvements have been overlooked.
>
> >In terms of generalization and robustness, our method has been rigorously evaluated on four datasets with significant domain differences: **KITTI, Make3D, NYUv2, and ScanNet**. These datasets cover a wide variety of scenarios, including outdoor driving scenes, reflective surfaces, low-texture regions, and complex indoor environments. Across all datasets, our method consistently outperforms existing self-supervised methods (as shown in **Tables 1-4 of the manuscript**) and baselines (please see **Table 5 in the manuscript**, and Table in **A4**) in both quantitative and qualitative metrics. This demonstrates our method’s robustness and generalization capability across diverse settings.
>
> >In summary, while reflective regions remain difficult for all existing methods, our approach has the ability to produce coherent global structures. More importantly, our method goes beyond this single challenge to tackle other significant issues, such as low-texture regions, thin structures, and shadow occlusions, resulting in a robust and generalizable solution. We sincerely appreciate your valuable feedback and hope this response adequately addresses your concerns. We also respectfully hope that you recognize the overall contributions and significance of our work.

---

> > ### Comment · Reviewer_ypGr · 2024-11-26
> > **Discussion on refective/low-texture regions**
> >
> > I believe the authors overclaim that the method can solve the reflective and low-texture regions. At least, from the visual examples, we can easily see many failed cases.
> >
> > I agree that the depth estimation for reflective areas is a challenging problem. Their method is better than the compared ones. However, such methods are too old. Maybe recent SOTA has solved this problem. If so, the paper's argument on these flaws.
> >
> > Furthermore, the paper argues they can solve the low-texture regions In the last row of Fig. 7, the cabinet is a low-texture example, but the method failed.

---

> > > ### Author Response · Authors · 2024-11-29
> > > **Response to disscussion**
> > >
> > > >Thank you for your patient feedback. We genuinely appreciate your thoughtful insights and emphasis on the imperfections in our results. However, we do not wish for our paper to be solely judged based on a handful of examples. We sincerely hope for a comprehensive evaluation that takes into account both the contributions and limitations of our work. We would like to highlight the following points.
> > >
> > > >**Overclaim:** We did not claim that our approach could completely solve the low-texture and reflective issue. Rather, in both the manuscript and our response, we claim that our approach can better address the challenging cases of depth estimation in real-world scenarios, which are majorly caused by lack of discriminative depth cues (*e.g.*, low-texture regions, reflective regions, *etc.*).  These challenges are long-standing in self-supervised monocular depth estimation, and many existing approaches are easily failed in those scenarios, however, in this paper, we have shown considerable progress in handling such challenging issues.
> > >
> > > >Specifically, our approach reduces the Sq Rel error metric by 10%, 4.4%, 9.5%, 4.9% on **four benchmark datasets** (KITTI, Make3D, NYUv2, and ScanNet), showcasing its robustness and generalization across diverse environments.
> > > Furthermore, our method has been tested against challenging scenarios (See **A1**), including reflective and low-texture regions, where it demonstrates greater robustness compared to existing approaches.
> > >
> > > >In addition, as you pointed out, our method is better than the compared ones in overall on previous presented results in general scenarios (see Fig. 3-7 of the Appendix) and challenging scenarios (see Fig. 9 of the Appendix). We observe that the recent works still obtain incomplete depth results in low-texture areas and reflective areas, as well as discontinuous depth results in shadow-occluded areas, while our method achieves better results in these challenging areas. We noted that our method also fails in some cases, but it is obviously better than existing methods in most cases. We would like to highlight that the visualization demonstrated in the paper were not cherry-picked, although existing papers might carefully pick the "best examples" as you mentioned. We have provided an additional qualitative comparison of our method with the recent SOTA work [1] in Fig. 14 of the manuscript. We sincerely hope for your understanding that the flaws in several examples do not outweigh the merits of our contributions. To support further research, we also discuss the limitations of our current method in **Section L of Appendix** and propose potential solutions.
> > >
> > > >**Contribution**: We believe that the core idea of this work would benefit the research of self-supervised monocular depth estimation. In this work, we propose a novel self-supervised monocular depth estimation framework based on the Three-dimensional Scene Field representation. Unlike previous methods that employ only the front-view 2D features, we design a 3D scene field to recover the multi-view representation and then capture sufficient structure- and orientation-aware 3D geometric features from it for robust depth estimation. Our approach utilizes an encoder-decoder architecture. The encoder extracts scene features from the input 2D image, which are then reshaped into a tri-plane feature field comprising three orthogonal, axis-aligned feature planes. By incorporating scene prior encoding, this tri-plane feature field effectively models the structure and appearance of the continuous 3D scene. The depth map is estimated by simulating the camera imaging process, which involves constructing a 2D feature map with 3D geometry by sampling from the tri-plane feature field.  The aggregated multi-view geometric feature map is then fed into the decoder for depth estimation. To the best of our knowledge, our TSF-Depth is the first work to model 3D scene filed for monocular depth estimation. As mentioned above,  extensive experiments on widely used outdoor datasets (KITTI and Make3D) and indoor datasets (NYUv2 and ScanNet) demonstrate the robustness and generalization-ability of our proposed approach.
> > >
> > > >[1] Sun, Yihong, and Bharath Hariharan. "Dynamo-depth: fixing unsupervised depth estimation for dynamical scenes." Advances in Neural Information Processing Systems 36 (2024)

---

> ### Author Response · Authors · 2024-11-25
> **Response to shift**
>
> >Thank you for raising this insightful concern regarding the potential impact of depth shift in our method. We would like to clarify that the shift is not critical in our approach due to the specific design of our framework, which inherently reduces the influence of shift. Firstly, our method employs a learnable tri-plane feature field, where the flexibility of the feature representation allows it to adapt to potential shifts in the point cloud. When the point cloud experiences a shift, the tri-plane feature field can dynamically adjust, ensuring that robust geometric features are sampled for each point. The joint learning between the point cloud and the feature field helps to mitigate the impact of potential shift, enabling the model to achieve stable and accurate depth predictions.  Furthermore, by combining multi-scale point clouds with multi-scale tri-plane feature fields, our method further enhances robustness to shift. Moreover, the multi-scale consistency loss ensures that depth predictions remain stable and geometrically consistent across different resolutions, effectively reducing the impact of global or local shift. These design choices eliminate the need for explicitly modeling or minimizing the shift during training.
>
> >Therefore, while depth shift is indeed a critical challenge in affine-invariant depth estimation, the specific design of our framework inherently mitigates its impact. Additionally, our quantitative and qualitative results on multiple datasets (**KITTI, Make3D, NYUv2, and ScanNet**), along with ablation studies on the KITTI and NYUv2 datasets, validate the effectiveness of our proposed method for depth estimation.
>
> >We sincerely thank your valuable feedback and hope this clarification addresses your concerns. Please do not hesitate to reach out if further details or analyses would be helpful.

---

> ### Author Response · Authors · 2024-11-25
> **Response to point cloud visualization**
>
> >Thank you for your constructive comment regarding the visualization of the reconstructed point clouds.  The visualizations of the reconstructed point clouds from various perspectives are presented in the **Fig. 12 of the manuscript**. We observed that under the three viewing perspectives, the overall structure of point clouds corresponding to different scales is roughly consistent. Although distortion is observed in some areas of the point cloud from the side view, as we responded to shift, the distortion of the point cloud does not affect the sampling of robust geometric features for depth estimation.

---

> ### Comment · Reviewer_ypGr · 2024-11-26
> **Discussion**
>
> Currently, I cannot conclude that the shift does not affect the depth and distortion. From the affine-invariant depth formulation, all previous works have demonstrated that the shift will cause distortion. Maybe the proposed method can dynamically adjust the shift. However, the paper does not theoretically prove this. Furthermore, the paper does not provide mutli-view reconstructed point clouds for visually evaluation. Authors could follow Marigold's teaser image for multi-view visualization.

---

> > ### Author Response · Authors · 2024-11-29
> > **Continued response to disscussion**
> >
> > >**Multi-view point cloud**: Multi-view visualization results are already included in Fig. 12 of the manuscript, as noted in our response to point cloud visualization. Additionally, we also report more multi-view visualization results in Fig. 13 of the manuscript. From these visualizations, we observe that the overall structure of the point clouds corresponding to different scales remains roughly consistent. Of course, distortion is also observed in some areas of the point cloud from the side view. Note that multi-view point clouds are reconstructed based on depth upsampling at various scales. Therefore, the smaller the scale, the greater the distortion present in the point cloud. However, as we discussed in response to shift, the distortion of the point cloud does not affect the sampling of robust geometric features for depth estimation.
> >
> > >**Comparison method:** In our qualitative comparison, we compare our method with SOTA approaches, including Lite-Mono-8M and StructDepth. As shown in Tables 1-4 of the manuscript, Lite-Mono-8M and StructDepth are already at the leading level.
> >
> > >Finally, we would like to express our sincere gratitude for your thoughtful review, and  deeply appreciate the opportunity to address your comment. This research has been a long and challenging journey for our team, and we are really excited about the learning 3D scene field for monocular depth estimation. We believe that the technical contributions and experimental results presented in this paper can make a meaningful impact on the field, offering new insights and advancements in monocular depth estimation. Our goal is to push the boundaries of what is possible and to contribute in a meaningful way to the ongoing development of this area.  Once again, thank you for your review. We deeply value your suggestions and hope for your reconsideration of this work.

---

> ### Author Response · Authors · 2024-12-03
> **Kind Reminder: Discussion Deadline Approaching**
>
> Dear Reviewer ypGr,
>
> We sincerely thank you for your valuable feedback and insightful suggestions. As the discussion period is coming to a close, we would greatly appreciate it if you could let us know whether our responses have addressed your concerns or if there are any additional points you would like us to clarify. If our responses have satisfactorily addressed your concerns, we kindly ask you to consider reflecting this in your score.
>
> Once again, thank you for your time and thoughtful input throughout this process.
>
> Best regards,
>
> Authors of submission 243

---

### Author Response · Authors · 2024-11-24
**Response to all reviewers**

We sincerely thank the reviewers for their time and effort in reviewing our manuscript and for their insightful and constructive feedback, which have significantly enhanced the quality of our work.
We greatly appreciate that the reviewers acknowledge the strengths of our work: "**The manuscript is well-structured and easy to follow** "(Reviewer ypGr), "**Clear contribution with tri-plane features**" (Reviewer kECm), "**The proposed method is novel to the best of my knowledge**" and "**The experiments are thorough**", and "**The proposed TSF-Depth achieves state-of-the-art performance**" (Reviewer Qiro).

We have revised the manuscript according to your suggestions, with all changes highlighted in cyan in the revised manuscript.
We have also responded to each of your concerns under your respective reviews. Below we summarize the responses to each reviewer’s comments.

>**In response to Reviewer ypGr**: We designed a challenging test subset to assess the robustness of our approach, and provided more qualitative comparisons. Furthermore, we included a mathematical analysis to illustrate how our approach mitigates the shift problem.
Additionally, we presented visualizations of reconstructed point clouds at different scales and reported quantitative results of the baseline in more datasets.

>**In response to Reviewer KECm**: We added all notations to Fig. 2 to improve the clarity of the manuscript and provided more qualitative results on the remaining three datasets.

>**In response to Reviewer nMyD**: We compared the depth improvement achieved when our approach uses same encoder as Lite-Mono.

>**In response to Reviewer Qiro**: We added the keyword "self-supervised" to the title and clarified the experimental setting of the ablation experiments.
We also answered the confusion about "3D representation capability" and "Adaptability to side views and cropped images".  Additionally, we experimentally verified the improvement of our method and reported visualization results for intermediate depth maps and point clouds.

We hope that our responses below convincingly address all reviewers’ concerns.  Please feel free to let us know if additional clarifications are needed. Again, we thank all reviewers for their time and efforts.

---

### Comment · Area_Chair_uXMC · 2024-11-27
**Reminder: Last day for author feedback**

This is a reminder that today is the last day allotted for author feedback. If there are any more last minute comments, please send them by today.

---

### Note · Authors · 2025-02-11

I have read and agree with the venue's withdrawal policy on behalf of myself and my co-authors.

---

### Meta-Review · Area_Chair_uXMC · 2024-12-20

**Metareview:**

The authors proposed a method for self-supervised monocular depth estimation. The main claims of novelty included a tri-plane feature field and a 3D-to-2D module to sample from the tri-plane representation. We have read the referee reports, the author responses. There were several points raised by the referees including the soundness of the proposed tri-plane representation (ypGr, Qiro), robustness of the proposed approach (ypGr), additional comparisons/extensions (kECm, Qiro). ypGr raised concerns regarding possible overclaim of the robustness of the approach. The authors provided additional experiments, but was unable to fully address this concern. We appreciate the additional experiments done by the authors, but feel that the authors can tone down regarding the claim. More critically are the algorithmic concerns raised by ypGr on distortion and Qiro on sensitivity to image size; both ypGr and Qiro did not feel their concerns were fully addressed.

Since the scores of this submission were on borderline (two 5s, two 6s), we read the manuscript and found that the proposed tri-plane representation bares many similarities as those in existing work from reconstruction [A] to generation [B, C], yet there was no mention of the previous work and claims the tri-plane representation (termed Three-dimensional Scene Field representation) as the main novelty. While we understand that much of computer vision involves building on top of existing methods and sharing ideas across problems, one must give credit to those who came before.

[A] Zou et al. Triplane Meets Gaussian Splatting: Fast and Generalizable Single-View 3D Reconstruction with Transformers. CVPR 2024.

[B] Chan et al. Efficient Geometry-aware 3D Generative Adversarial Networks. CVPR 2022.

[C] An et al. PanoHead: Geometry-Aware 3D Full-Head Synthesis in 360. CVPR 2023.

Additionally, as baselines and comparisons were a point of contention, we went over the experiment results and found that the more recent methods selected for comparisons were either NOT the top methods published at top computer vision conference [E, F, G, I, J], not peered reviewed [H], or were focused largely on a different setting such as dynamic scenes [D]. Based on what we have observed in performance trends of self-supervised monocular depth estimation methods, those published in computer vision conferences such as WACV [K], ICCV [L], ECCV [M] were much more competitive and beat the proposed method across all metrics. For example, on KITTI under the monocular training setting using 640 ×192 resolution, we have listed the results of TSF-Depth (proposed, taken from Table 1) and those of [K, L, M]:

| Method | Sq Rel ↓ | Abs Rel ↓ | RMSE ↓ | RMSE log ↓ | δ<1.25 ↑ | δ<1.252 ↑ | δ<1.253 ↑ |
|-----|-----|-----|-----|-----|-----|-----|-----|
| TSF-Depth (proposed) | 0.692 | 0.096 | 4.335 | 0.173 |  0.903 | 0.967 | 0.984 |
| [K] (WACV 2023) | 0.665 | **0.093** | 4.272 | 0.172 | **0.907** | 0.967 |  0.984 |
| [L] (ICCV 2023) | 0.661 | 0.099 | 4.316 | 0.173 | 0.897 | 0.967 | **0.985** |
| [M] (ECCV 2022) | **0.632** | 0.096 | **4.216** | **0.171** | 0.903 |  **0.968** | **0.985** |

We understand that it is common for literature to be missed. However, we caution the authors in their selection of comparisons and making the overclaim to be the state of the art. We urge the authors to consider a literature dive and report the results of existing competitive methods.



[D] Sun et al. Dynamo-depth: Fixing unsupervised depth estimation for dynamical scenes. NeurIPS 2023.

[E] Zhao et al. Learning effective geometry representation from videos for self-supervised monocular depth estimation. ISPRS
International Journal of Geo-Information, 2024.

[F] Xiong et al. Monocular depth estimation using self-supervised learning with more effective geometric constraints. Engineering
Applications of Artificial Intelligence, 2024.

[G] Feng et al. Shufflemono: Rethinking lightweight network for self-supervised monocular depth estimation. Journal of Artificial
Intelligence and Soft Computing Research, 2024.

[H] Liu et al. Unsupervised monocular depth estimation based on hierarchical feature-guided diffusion. arXiv preprint, 2024.

[I] Guo et al. Simmultidepth: Self-supervised indoor monocular multi-frame depth estimation based on textureaware masking. Remote Sensing, 2024.

[J] Guo et al. F2depth: Self supervised indoor monocular depth estimation via optical flow consistency and feature map synthesis. Engineering Applications of Artificial Intelligence, 2024.

[K] Chen et al. Self-Supervised Monocular Depth Estimation: Solving the Edge-Fattening Problem. WACV 2023.

[L] Han et al. Self-Supervised Monocular Depth Estimation by Direction-aware Cumulative Convolution Network. ICCV 2023.

[M] He et al. RA-Depth: Resolution Adaptive Self-Supervised Monocular Depth Estimation. ECCV 2022.

**Additional Comments On Reviewer Discussion:**

There were several points raised by the referees including the soundness of the proposed tri-plane representation (ypGr, Qiro), robustness of the proposed approach (ypGr), additional comparisons/extensions (kECm, Qiro). ypGr raised concerns regarding possible overclaim of the robustness of the approach. The authors provided additional experiments, but was unable to fully address this concern. We appreciate the additional experiments done by the authors, but feel that the authors can tone down the text regarding the claim. More critically are the algorithmic concerns raised by ypGr on distortion and Qiro on sensitivity to image size; both ypGr and Qiro did not feel their concerns were fully addressed.

Nonetheless, because of the borderline decision, we have read the manuscript and found several critical concerns including the overclaims of novelty on the idea of tri-plane representation, without citing previous works which it originates. Additionally, there were false claims of being state of the art with omission of existing methods that were better than the proposed method.

---

### Decision · Program_Chairs · 2025-01-22

Reject